# Does Precision-Based Medicine Hold the Promise of a New Approach to Predicting and Treating Spontaneous Preterm Birth?

Hiba Khan [1], Natasha Singh [2], Luis Yovera Leyva [3], Johann Malawana [1] and Nishel M. Shah [2,*]

1 School of Medicine, University of Central Lancashire, 135A Adelphi Street, Preston PR1 7BH, UK; hiba.khan7@nhs.net (H.K.); jmalawna@uclan.ac.uk (J.M.)
2 Department of Metabolism, Digestion and Reproduction, Imperial College London, Chelsea and Westminster Hospital, London SW10 9NH, UK; n.mohammed@imperial.ac.uk
3 Department for Women and Children, Broomfield Hospital, Chelmsford CM1 7ET, UK; luis.yovera@nhs.net
* Correspondence: nishel.shah@imperial.ac.uk

**Abstract:** Background: Preterm birth (PTB) is a leading cause of childhood disability, and it has become a key public health priority recognized by the World Health Organization and the United Nations. Objectives: This review will: (1) summarize current practice in the diagnosis and management of PTB, (2) outline developments in precision-based medicine for diagnostics to improve the care provided to pregnant women at risk of PTB, and (3) discuss the implications of current research in personalized medicine and the potential of future advances to influence the clinical care of women at risk of PTB. Methodology: This is a narrative literature review. Relevant journal articles were identified following searches of computerized databases. Key Results: Current and emerging technologies for the utility of personalized medicine in the context of PTB have the potential for applications in: (1) direct diagnostics to identify and target infection as one of the main known causes of PTB, (2) identifying novel maternal and fetal biomarkers, (3) the use of artificial intelligence and computational modeling, and (4) combining methods to enhance diagnosis and treatment. Conclusions: In this paper, we show how current research has moved in the direction of the targeted use of biomarkers in the context of PTB, with many novel approaches.

**Keywords:** pregnancy; preterm birth; preterm labor; personalized medicine; precision-based medicine; biomarkers; genomics; artificial intelligence; diagnostics; immunomodulation

## 1. Introduction

Preterm birth (PTB), defined as neonates born before the 37th week of gestation, is a leading cause of death and disability in children under five years worldwide. Globally, there are 15 million PTBs per year [1]. As such, PTB is a key public health priority that is recognized and closely monitored by the World Health Organization (WHO). In the UK, 52,000 babies a year are born preterm, of which 8000 are born before 32 weeks [2]. These babies are disproportionately affected by significant long-term morbidity when compared to births at later gestations. For women identified as high-risk in early pregnancy, prevention strategies can be initiated to modify their risk. However, in more than 50% of women who deliver preterm, there are no identifiable risk factors to target [3]. Worse still, when these women present to hospital with early signs of preterm labor (PTL), there is no effective treatment to delay their progression to birth. Furthermore, over the last decade, there has been no measurable improvement in PTB rates despite changes in national and international guidance and improvements in the education of healthcare professionals [4]. Therefore, clinical care is aimed at preparing the neonate for PTB using interventions such as maternal steroids for fetal lung maturity and magnesium sulphate to reduce the risk of cerebral palsy [5]. In the United Kingdom (UK), the Government's 'Safer Maternity Care' action plan set a target of reducing the PTB rate from 8% to 6% by 2025 and included subsequent tributary initiatives such as the 'Saving Babies Lives Bundle 2',

PERIprem (Perinatal Excellence to Reduce Injury in Premature Birth) initiative and 'Better Births' vision to equip healthcare professionals with the tools they need to predict, suspect, diagnose and delay PTB as quickly as possible [6]. However, the PTB rate remains at 8% in the UK, and one of the key recommendations from the 'Safer Maternity Care' Progress report was the enabling of innovation in local clinical practice [7]. However, it has been recognized by stakeholders that PTL is a complex problem with several different etiologies, meaning that for any approach to the management of PTL to be effective, the first step must be to identify the cause in each individual woman [8]. Personalized medicine, often referred to as precision-based medicine, has been developing in many areas of medicine, most recently in the fields and research related to respiratory disease, hematology and cancer care. Contrary to the population approach, it offers an individualized targeted approach that could be ideal in the context of the early identification and delay of PTB [9].

In this narrative review, we performed a scoping literature search of computerized databases (PubMed, Scopus MEDLINE, EMCARE, EMBASE and CINAHL), limiting our search to English language papers. We identified relevant abstracts and articles and used these as a basis to describe current practice in the diagnosis and management of PTL and explored both historic and current research (conducted between December 1991 and October 2023), investigating novel approaches to diagnostics using precision-based medicine techniques.

## 2. Preterm Birth

PTB is divided into the following sub-categories: extremely preterm (less than 28 weeks), very preterm (28 to 32 weeks) and moderate to late preterm (32 to 37 weeks). The majority of PTBs are moderate to late, which account for 85%, and the remainder are split between extreme and very preterm, with rates of 4% and 11%, respectively [1]. The morbidity and mortality associated with PTB are considerable. Moreover, the earlier the delivery, the higher the risk of disability or death [10]. Neonatal complications of PTB include chronic lung disease, developmental delay, growth reduction, hearing impairment, intraventricular hemorrhage, necrotizing enterocolitis, nosocomial infections, patent ductus arteriosus, periventricular leukomalacia, respiratory distress syndrome, retinopathy of prematurity and pulmonary barotrauma [11].

For those babies who are born preterm, survival has improved significantly, with better neonatal care. As a result, much earlier gestations surviving long-term. In addition, the global incidence of PTB has decreased by 5.26%, from 16.06 million in 1990 to 15.22 million in 2019. However, deaths decreased much more significantly, by 47.71%, from 1.27 million in 1990 to 0.66 million in 2019 [12]. This is in part due to interventions such as fetal monitoring, standardized preterm care protocols, delayed cord clamping, deployment of evidence-based training across the multidisciplinary team and increased awareness from the results of the national MMBRACE (Mothers and Babies: Reducing Risk through Audit and Confidential Enquiries) perinatal surveillance program [13].

An example of one of the key outcomes of this work was PERIprem. PERIprem (Perinatal Excellence to Reduce Injury in Premature Birth) is an 11-element care bundle that was launched in May 2020 to deliver timely care to women at risk of preterm birth and improve outcomes [14]. The bundle comprises many of the elements discussed above, including birth in the right place, antenatal steroids, magnesium sulphate, optimal cord management, thermoregulation, maternal early breast milk, volume-targeted ventilation, caffeine, prophylactic hydrocortisone, probiotics and intrapartum antibiotics, and it was shown to significantly reduce neonatal morbidity and mortality in the regions implemented.

### 2.1. Risk Factors and Pathophysiology of Preterm Birth

There are several risk factors for PTB that can be identified preconception and in early pregnancy (Table 1). These can be divided into modifiable (such as smoking) and non-modifiable risk factors (such as previous cervical surgery or PTB) to direct preventative interventions or methods of surveillance.

**Table 1.** Table to show the recognized risk factors for preterm birth, categorized as past medical history, lifestyle, pregnancy complications and other.

| Past Medical History | Pregnancy Complications |
|---|---|
| Previous preterm birth | |
| Short cervix < 25 mm | Carrying more than one fetus |
| Early cervical dilatation | Vaginal bleeding during pregnancy |
| Past procedures on the cervix (LLETZ) | Infections during pregnancy |
| Injury during a past delivery | |

| Lifestyle | Other |
|---|---|
| Low pre-pregnancy weight | |
| Smoking during pregnancy | |
| Dietary deficiencies | Younger than 17 or older than 35 years |
| Injury during a past delivery | |

Royal College of Obstetricians and Gynecologists.

Though the exact pathophysiology of PTB remains unclear, most researchers agree that it is likely to be multifactorial (Figure 1). However, the most prevalently recognized mechanism is an inflammatory process occurring within the amniotic cavity due to maternal infection [15]. This may occur in around 40% of cases of PTL before 34 weeks, as a result of a microbial invasion of the amniotic cavity (MIAC) that precedes labor [16]. Other potential causes of PTB include a disruption of the vaginal microbiome that is linked with infection- and inflammation-mediated PTB, modified glycan-binding protein circuits that cause a disruption in maternal immune tolerance of the semi-allogenic fetus and maternal psychological stress manifesting through a neuroendocrine response [17–20].

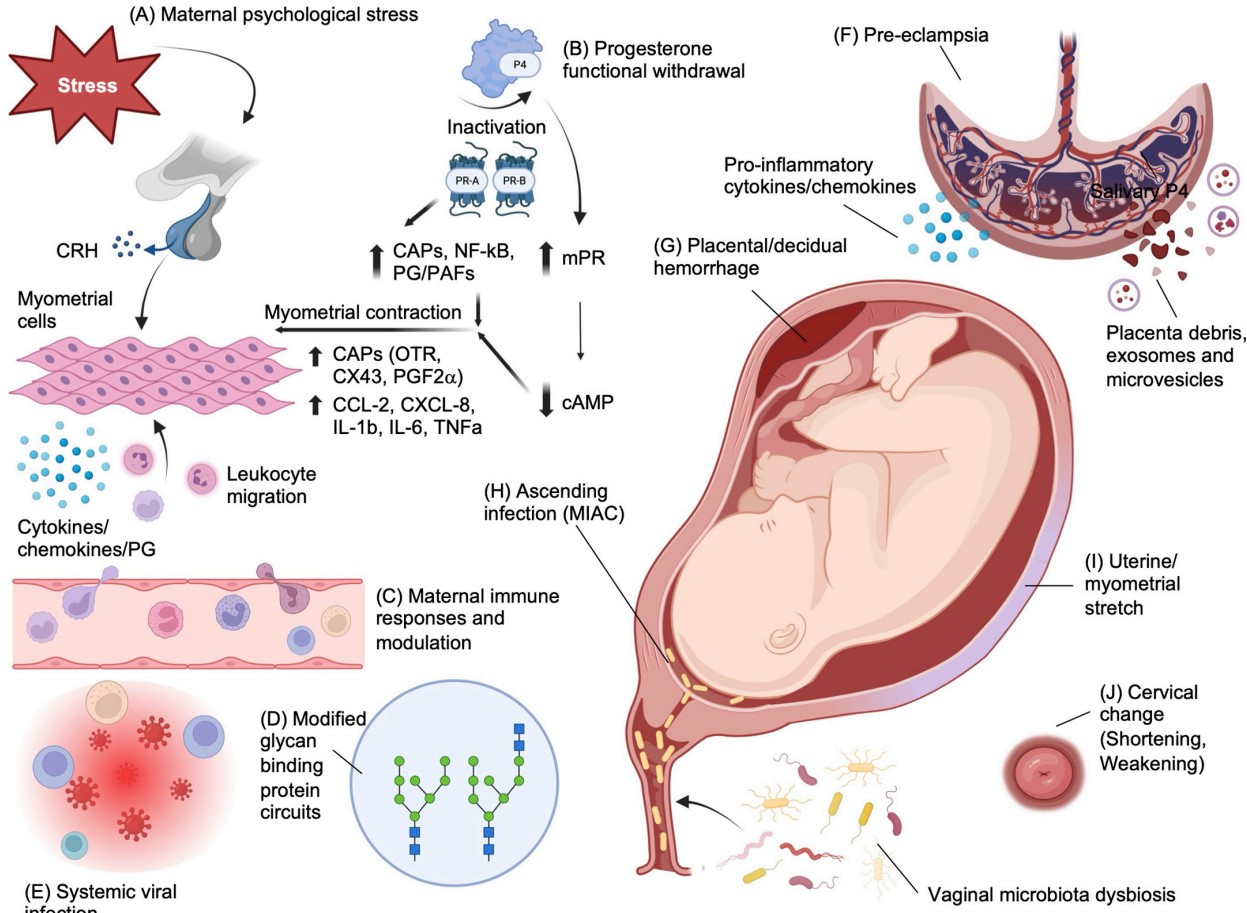

**Figure 1.** Pathophysiological mechanisms of preterm birth. (**A**) In contrast to the inhibitory effects of cortisol on hypothalamic CRH secretion, maternal stress causing cortisol release from the zona

fasciculata of the adrenal cortex can increase CRH production by the placenta and may provide a fetal trigger for labor by upregulating cortisol and dehydroepiandrosterone sulphate (DHEAS) release by the fetal adrenals. DHEAS is metabolized in the placenta to estrogens, which results in upregulation of proinflammatory cytokines and chemokines that mediate myometrial cell contractility. (**B**) Functional progesterone withdrawal due to the differential expression of nuclear progesterone receptor (PR) isoforms PR-A and PR-B in myometrial smooth muscle cells, favoring a greater PR-A/PR-B ratio prior to labor onset that upregulates pro-contractile gene expression. The overall effect is an upregulation of contractile-associated proteins (CAPs), including OTR, COX-2, Cx43 and prostaglandin F2α (PGF2α), and upregulation of pro-inflammatory cytokines. In addition to this, membrane PRs (mPRs) mediate progesterone intracellular signaling through activation of G-proteins and subsequent upregulation of adenylyl cyclase activity that decreases cAMP levels, which causes an increase in myosin phosphorylation. All this is further enhanced by a background of functional progesterone withdrawal and a reduction in PR-B transcriptional activity. (**C**) Dysregulated maternal immune responses due to (**D**) modified glycan-binding protein circuits that are responsible for several immune-modulatory functions, such as those mediated by progesterone as well as antimicrobial activity, and dendritic cell and NK interactions with trophoblast during placentation; and to (**E**) chronic viral infections (such as HIV) causing pro-inflammatory immune responses. Together, this disruption in immune regulation causes inflammatory triggers that can induce myometrial contractility via NFκB. (**F**) Pre-eclampsia is a known risk factor for iatrogenic and spontaneous PTB through a number of mechanisms that include abnormal trophoblast invasion during placentation (e.g., via the PI3K-AKT pathway) and release of syncytiotrophoblast debris and placental exosomes and microvesicles that predispose to maternal endothelial dysfunction caused by circulating placental-derived sFlt and an altered local and systemic maternal immune response that favors TH1 cytokines. (**G**) Decidual hemorrhage presenting as vaginal bleeding or retroplacental hematoma results in formation of thrombin that can stimulate myometrial contractility but also inhibits progesterone and mediates infiltration of neutrophils to the area. (**H**) MIAC through ascending organisms, including those associated with vaginal dysbiosis, causes intra-amniotic infection that can induce inflammation of feto-maternal tissues and immune dysregulation and myometrial contractility. (**I**) Uterine distension due to polyhydramnios, multiple pregnancies and large-for-gestational-age fetuses result in a myometrial stretch response with upregulated IL-8, as well as COX-1/2 activity. (**J**) Cervical pathology includes congenital factors and conditions such as Ehlers Danlos, Marfan's syndrome and cervical agenesis or dysgenesis that develop as a consequence of congenital collagenopathies. Acquired conditions include cervicitis caused by sexually transmitted infections or non-sexually acquired infections that ascend from the vagina to the cervix. The cervical epithelium can also be weakened by mechanical or chemical trauma and previous surgeries such as cone and excision biopsies to treat cervical intraepithelial neoplasia (CIN) and prevent cancer of the cervix. Created with Biorender.com—accessed on 1 November 2023.

A common neuroendocrine mechanism associated with spontaneous PTB is the elevation of placental corticotrophin-releasing hormone (CRH) levels due to a fall in CRH-binding protein (CRHBP), leading to presumed cervical ripening and a myometrial contraction response [21]. The theory of functional progesterone withdrawal reasons that the effect of continued high levels of progesterone throughout pregnancy and labor on myometrial tissue is attenuated due to a regulated metabolic progesterone withdrawal. This occurs through the differential expression of progesterone receptor isoforms, reduced activation and expression of progesterone co-activators, binding of progesterone to receptors that activate alternative cellular pathways, direct progesterone withdrawal mediated by the upregulation of cytokines and chemokines, and catabolism of progesterone from an active to an inactive compound in the uterus [22,23]. Furthermore, downregulation of placental progesterone has been shown to be related to an upregulation of placental CRH, which could lead to spontaneous PTB [21]. As an immune-modulator, supplementation with and antagonism of progesterone has been shown to have anti-inflammatory and pro-inflammatory effects ex vivo [24].

A common cause of iatrogenic PTB includes placenta previa, accreta, vasa previa and velamentous insertion of the cord [25]. These abnormalities of placental development are associated with significant maternal-fetal morbidity and mortality that, in the cases of previa, accreta and vasa previa, are not compatible with vaginal birth and so are delivered before 37 weeks of gestation.

Another iatrogenic cause is uteroplacental insufficiency, which is very often discussed in relation to pre-eclampsia (PET). The risk of PTB associated with PET is based on disease severity and often co-relates with the extent of uteroplacental insufficiency and consequent fetal growth restriction. The extent of placental insufficiency is believed to be due to reduced placental perfusion mediated through the vasoconstricting effects of the renin-angiotensin pathway, leading to sub-optimal trophoblast invasion [26]. It has been recognized that 30% of patients with PTL have been found to have placental lesions [27]. The mechanism behind vaginal bleeding or retroplacental hematoma formation is believed to be due to a defect in decidual hemostasis, leading to decidual hemorrhage or placental abruption. This triggers thrombin-stimulated myometrial contractility through protease-activated receptors PAR1 and PAR3 as well as degradation of the extracellular matrix, causing chorioamniotic membrane compromise [28]. Aside from PET, at a microscopic level, shallow invasion of the decidua, umbilical chorionic vasculitis and villitis and other decidual vascular abnormalities are also placental findings associated with infective PTB [29,30]. Though inflammatory pathways are involved in other forms of spontaneous PTL, they do not always result in histological changes in placenta. One example is in uterine stretch, where myometrial and placental membranes do not show evidence of tissue injury or remodeling [31].

Uterine overdistension is another mechanism for PTB. This can be due to polyhydramnios, multiple pregnancies, large-for-date pregnancies and the biomechanics of disproportionate maternal stature [32]. In vitro studies on human and rat myometrial cells have shown that mechanical stretching of the myocytes in this way leads to upregulation of oxytocin receptors, prostaglandins and gap junction proteins Cx43 and Cx26, which are associated with uterine contraction [33,34]. Waldorf et al. used a pregnant non-human primate model to show that uterine overdistension (using a uterine balloon catheter) results in the production of the pro-inflammatory cytokines IL-1b, TNF-$\alpha$, IL-6, IL-8 and CCL2 and induced a prostaglandin response in the amniotic fluid [31]. Peak levels of cytokines and prostaglandins correlated with the extent of uterine distention. Cytokine upregulation was also seen in vitro following stretch using human amniocytes (IL1$\beta$, IL6 and IL8 mRNA) and ex vivo in patients with polyhydramnios (TNF-$\alpha$ and IL-6) and twins (TNF-$\alpha$) [31].

Cervical insufficiency is also a recognized cause of PTL and is either congenital or acquired. Acquired causes of cervical insufficiency include cervical shortening through a procedure involving large loop excision of the transformation zone (LLETZ), cone biopsy, cervical lacerations from childbirth or weakening of the epithelium through laser ablation. The composition of the cervix comprises 60% type I fibrillar collagen, 30% type III fibrillar collagen and the proteoglycans hyaluronic acid, chondroitin sulphate, keratan sulphate and dermatan sulphate. As such, congenital conditions resulting from mutations in the genes that mediate the production of those types of collagens predispose to PTL. These include conditions such as osteogenesis imperfecta, Ehlers–Danlos syndrome and Marfan's syndrome [35]. The consequence of cervical insufficiency is the loss of biomechanical integrity to maintain fetal intrauterine support due to a more open internal cervical os as well as a loss of integrity of the cervical epithelial barrier, resulting in a higher risk of ascending bacterial infection, both of which can lead to PTB [36].

Chronic systemic infections such as human immunodeficiency virus (HIV) that cause altered maternal immune responses are also associated with PTB. Globally, it is estimated that there are over 19.3 million women of reproductive age with HIV [37]. Pregnant women with HIV have a three-fold increased risk of spontaneous PTB versus the background population risk [38,39]. The mechanism behind this is thought to be due to the dysregulation of CD4 and CD8 cell immune responses. This is in part mediated by a reduction in CD4

T cells and an altered phenotype of monocytes following viral infection. This results in an upregulation of a subset of suppressive Tregs (T-follicular regulatory cells), increased expression of the co-inhibitory receptor TIGIT on natural killer (NK) cells that attenuates their functional capacity and disturbances in the Th17/Treg ratio [40]. Even with treatment and subsequent immune reconstitution of CD4 T cells, the effects of chronic viral infection potentiate the proinflammatory environment, and some elements of immune dysregulation persist. Specifically, in pregnancy, the result is an upregulation of cytotoxic decidual NK cells and both decidual and peripheral CD8 T cells and an Increase in the proinflammatory subsets of (M1) monocytes and macrophages with cytokine (IL-6, TNFα and IL-1b) and phagocytic immune functions [41,42]. Moreover, both decidual and peripheral T helper subsets are altered, but specifically, at the maternal–fetal interface, there is a skewed T helper cell profile that results in further proinflammatory cytokine release [41].

In addition to immune dysregulation, several studies have implicated protease inhibitors used to treat HIV infection as a cause of PTB [43]. In 1998, Lorenzi et al. investigated the hypothesis that pregnant women taking reverse transcriptase and protease inhibitors were at higher risk of PTB to evaluate the safety of this HIV treatment for this demographic group and discovered a high incidence of prematurity (spontaneous PTB) in these women [44]. Two years later, a larger prospective study of 3920 European women was published that also found an association between protease inhibitors and PTB (17% PTB rate) [45]. However, no clear mechanism for this has been identified, and further work refutes this assertion [46]. Patel et al. subsequently performed a study of 777 pregnant women with HIV at Harvard and found no association between the use of protease inhibitors and PTB [47]. This was corroborated by work conducted by Williams et al. on 129 pregnant women in New Jersey that compared surrogate markers of antiretroviral therapy/HIV disease progression (CD4 counts and viral load) and birth outcomes between women taking protease inhibitors and those not taking protease inhibitors [48]. They found an increased risk of PTB in the protease inhibitor-treated group ($p$ = 0.018). More recently, in 2018, a UK-based study examining 4184 pregnant women with HIV found an association between PTB and protease inhibitors, in particular lopinavir [49]; and most recently, a review published in the *Lancet* concluded that protease inhibitors "might" contribute to uteroplacental pathology [50]. Though this topic remains a subject of debate, current advice is to prioritize the wellbeing of the pregnant woman living with HIV and her fetus, by ensuring that her anti-retroviral treatment maximally suppresses viral replication as early as possible to minimize vertical transmission of HIV. Nonetheless, treatment options to reduce the risk of PTB in this subgroup are an important area of research. The ProSPAR study, a multicenter randomized controlled trial investigating progesterone supplementation in HIV-positive pregnant women on protease inhibitor-based regimens, has been designed to explore the use of progesterone to counteract the potential increased risk of spontaneous PTB in these women, but is not yet underway [51]. Impaired vascular placental development and poor perfusion and placental insufficiency have also been observed in pregnant women with HIV [41]. Moreover, patients on anti-retroviral treatment have been shown to have lower serum progesterone levels, and this is a possible contributor to placental pathology and cause for protease inhibitor-associated PTB [52,53]. However, more recently, studies have revealed that the risk of PTB in women with HIV is lower with respect to how long the woman has been on antiretroviral therapy, with most women delivering at term [39].

The interplay and complexity of these causes are indicative of the multifactorial etiology of PTB. The National Institute for Health and Care Excellence (NICE) based in the UK and the WHO have published guidelines to support clinical staff to better identify risk factors for PTB in early pregnancy and offer evidence-based interventions [54,55]. In addition to recommending methods to modify risk, as well as methods of surveillance of high-risk groups, they also provide guidance on the nature of obstetric and neonatal support that should be available to patients in the acute setting where PTB is inevitable. These are outlined in Figure 2, Boxes 1 and 2. Importantly, these measures should be chosen after a multidisciplinary team discussion and jointly with the patient. Factors such as

gestation and an assessment of fetal wellbeing, co-existing maternal and fetal comorbidities, the level of neonatal care available, patient preferences and resource availability should all be considered when planning clinical care.

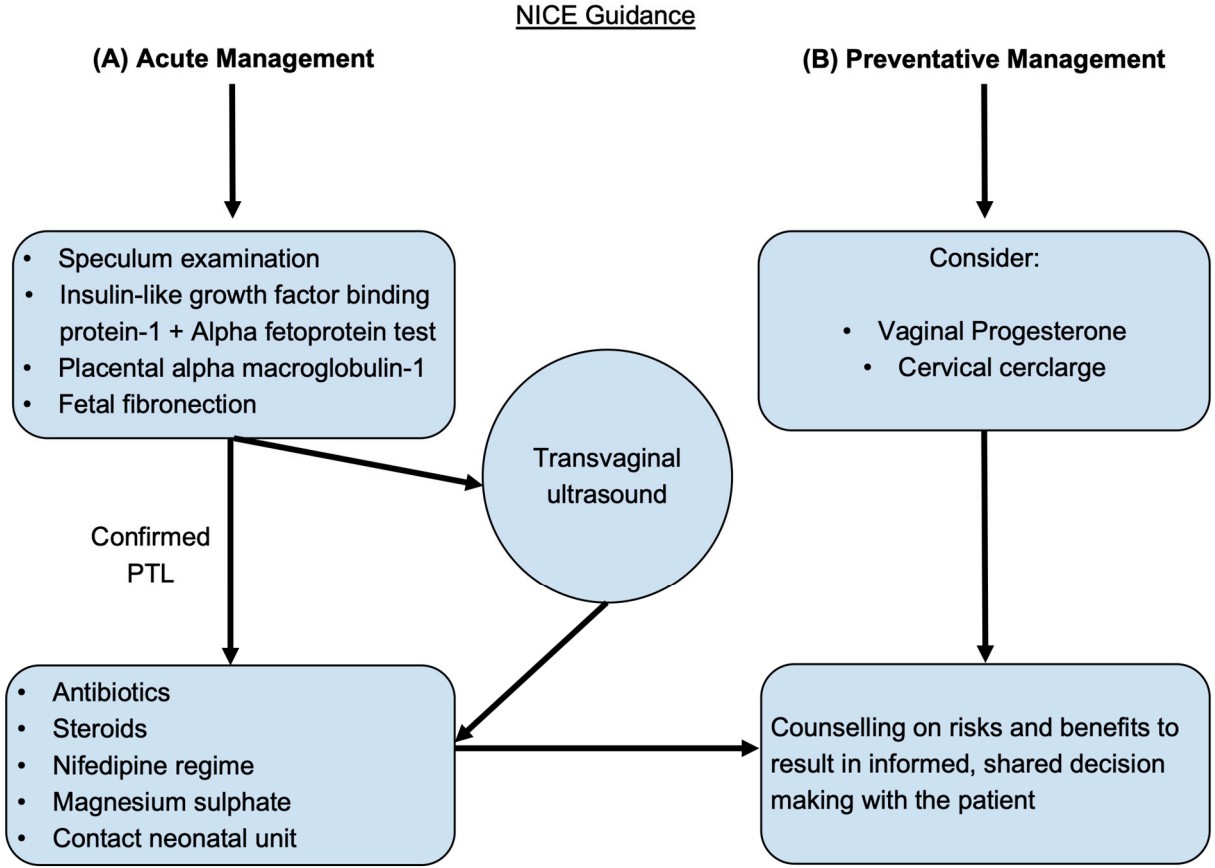

**Figure 2.** The National Institute for Health and Care Excellence (NICE) guidance for managing preterm labor. The flow chart shows current recommendations for (**A**) the diagnosis and management strategies for acute preterm labor (PTL) and for (**B**) the preventative treatments available for high-risk patients.

**Box 1.** Guidance for the management of women presenting with clinical evidence of preterm labor (PTL) and pre-labor preterm rupture of membranes (P-PROM). World Health Organization (WHO) recommendations (2022).

1. Offer antenatal corticosteroids for women with a high likelihood of preterm birth (PTB) from 24 weeks to 34 weeks of gestation when the following conditions are met:
    a. Gestational age assessment can be accurately undertaken.
    b. There is a high likelihood of preterm birth within 7 days of starting therapy.
    c. There is no clinical evidence of maternal infection.
    d. Adequate childbirth care is available (including the capacity to recognize and safely manage PTL and PTB).
    e. The preterm newborn can receive adequate care (including resuscitation, kangaroo mother care, thermal care, feeding support, infection treatment and respiratory support including continuous positive airway pressure (CPAP) as needed).
2. Offer tocolysis in the form of nifedipine for acute and maintenance therapy for women with a high likelihood of PTB.

**Box 2.** Guidance for the management of women presenting with clinical evidence of preterm labor (PTL) and pre-labor preterm rupture of membranes (P-PROM). Royal College of Obstetricians and Gynecologists (RCOG) and National Institute for Health and Care Excellence (NICE) recommendations (2022).

---

1. For women at risk of PTL, vaginal progesterone and prophylactic cervical cerclage can be considered.
2. To diagnose preterm, prelabor rupture of membranes (P-PROM), a speculum examination to look for pooling of amniotic fluid and an immunochromatographic binary point-of-care test to analyze amniotic fluid components such as insulin-like growth factor-binding protein-1 (ROMplus) or placental alpha macroglobulin-1 (PartoSure) if speculum examination is inconclusive. ROM plus is an immunochromatographic binary point-of-care test to identify two proteins found in amniotic fluid—insulin-like growth factor-binding protein-1 (IGFBP1) and alpha fetoprotein (aFP)—to determine rupture of fetal membranes. PartoSure helps to detect PTL in women with intact membranes through the identification of placental alpha macroglobulin-1 in the vaginal secretions of pregnant women.
3. Transvaginal ultrasound to diagnose cervical competency and the ability to carry a fetus to term.
4. If diagnosed with P-PROM, antenatal prophylactic antibiotics in the form of oral erythromycin QDS for 10 days or until established labor should be prescribed, and antenatal corticosteroids should be offered with counseling on the risks and benefits to both mother and baby.
5. Emergency cervical cerclage can be considered for women between 16 + 0 and 27 + 6 weeks of gestation who have intact membranes and no uterine activity, signs of infection or vaginal bleeding.
6. If membranes are intact, the use of fetal fibronectin for an understanding of delivery probability within the next 48 h.
7. Nifedipine is the drug of choice for tocolysis in suspected or diagnosed PTL with intact membranes from 24 + 0 to 33 + 6 weeks of gestation.
8. Women and family members should be counseled on the risks and benefits of antenatal corticosteroids, and steroids should be considered if the woman is in PTL from 22 + 0 to 35 + 6 weeks of gestation.
9. The use of intravenous magnesium sulphate should also be considered for neuroprotection of the child and offered to women between 24 + 0 and 33 + 6 weeks of gestation who are planning to have or are already in PTL.

---

## 2.2. Current Clinical Tools to Predict Preterm Birth

There are currently several aids available in clinical practice to predict PTB in high-risk groups. These include ultrasound, serum biomarkers, vaginal biomarkers and a clinical decision-making app (Figure 3). The Placental Growth Factor (PLGF) biomarker is also approved for use by the NHS and is available in a few units as a point-of-care test, primarily to be used to exclude pre-eclampsia. However, it is also a reliable predictor of disease progression and time to delivery [56]. As a result, it is used to predict the need for delivery within 14 days, with a sensitivity (the ability of the test to identify true positives) and specificity (the ability of the test to identify true negatives) of 80% for both and positive and negative predictive values (the proportion of individuals with a positive test result who will deliver preterm or with a negative test result who will not deliver preterm) of 0.41 and 0.95, respectively (using the Triage MeterPro point-of-care analyzer that uses a fluorescence immunoassay) [57]. More recently, PLGF has been investigated as a potential biomarker for predicting spontaneous PTL, and this is discussed later in this review [58].

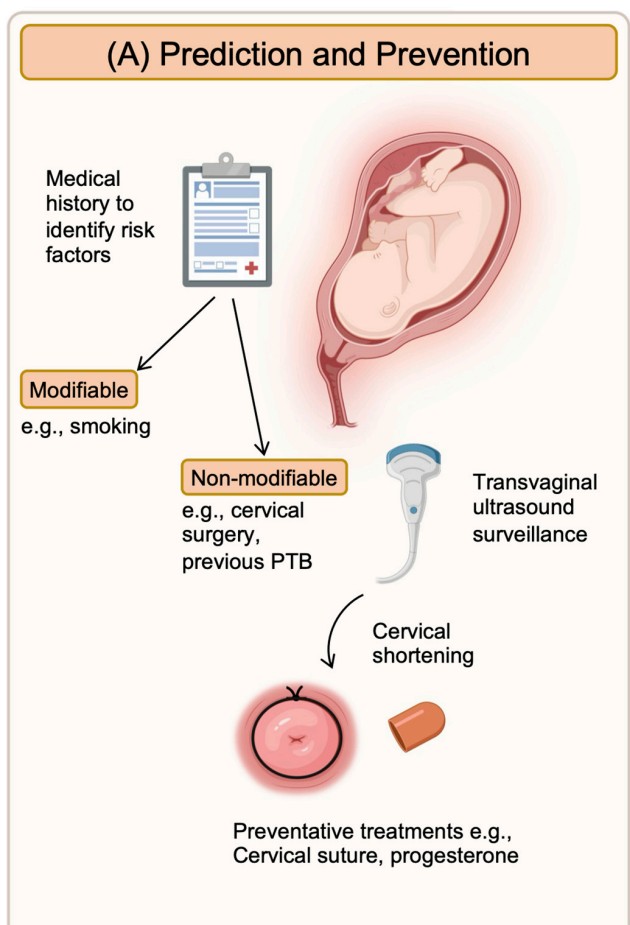
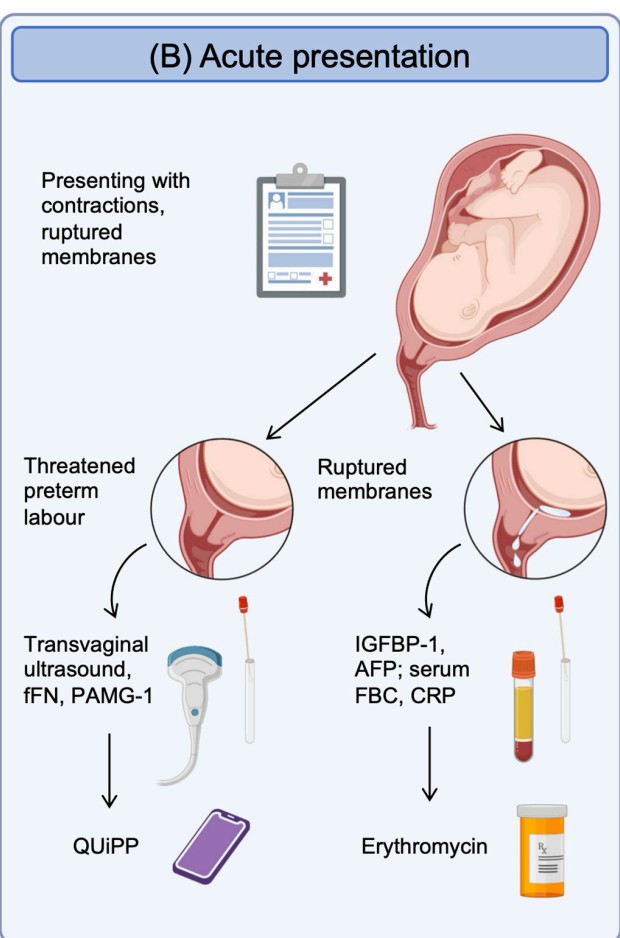

**Figure 3.** Current clinical tools available to help clinicians predict and manage preterm birth. (**A**) Prediction and prevention rely on identifying high-risk patients using their medical and obstetric history to determine risk factors. These can then be categorized into modifiable and non-modifiable groups. Those that are modifiable can be managed to reduce the individual's risk, such as smoking with smoking cessation. For those risks that are non-modifiable, surveillance strategies such as serial transvaginal ultrasound can be initiated. When cervical shortening occurs, preventative treatments can be started (e.g., cervical suture, progesterone supplements). (**B**) Acute presentations will be in the form of pregnant women with contractions and/or ruptured membranes. Confirmation will be with clinical examination, but diagnostic tools that will help include: (1) for threatened PTL, ultrasound (intact membranes) to assess cervical shortening and a vaginal swab to measure fFN or PAMG-1, which is released in labor from the cervix; (2) for ruptured membranes, the presence of proteins from the amniotic fluid (AFP and IGFBP-1) released into the vagina and serum markers of infection (full blood counts/FBC and CRP). New tools that combine methods to improve accuracy of these clinical tools include the QUiPP app, which combines cervical length with quantitative fFN concentrations. Created with BioRender.com—accessed on 13 December 2023.

Specifically for spontaneous PTB, ultrasound and fetal fibronectin (FFN) are used in clinical practice. The International Society of Ultrasound in Obstetrics and Gynecology Standards Committee (ISUOG) has published good practice recommendations with regards to the standardized and accurate measurement of cervical length, including measurement cut-offs (25 mm at <24 weeks has a 30% likelihood of spontaneous preterm delivery), best practice, operator qualifications and screening. Supportive indicators such as amniotic sludge were also recently reported by the ISUOG as an ultrasound finding that demonstrated increased risk of PTB [59]. Though transvaginal ultrasound to measure the cervix is primarily used for surveillance in high-risk groups, it can be used in the acute setting

to determine the risk of PTB within 2 weeks. In addition to ultrasound, a quantitative point-of-care test to measure cervicovaginal FFN provides an additional method to predict spontaneous PTB in symptomatic pregnant women. In a metanalysis of 64 studies with a total of 26,876 participating pregnant women, Honest et al. reported a likelihood ratio of >6 in four of the highest-quality studies for predicting PTB within 7–10 days [60].

Researchers from King's College London have developed the QUIPP app clinical decision-making tool, which combines cervical length and FFN with an individualized risk assessment to determine the likelihood of PTB in symptomatic women [61–64]. The app is highly predictive of delivery. It has a sensitivity 97.5% and a negative predictive value of 100% for predicting spontaneous PTB within the following 7 days [65]. Moreover, the area under the receiver operating characteristic curve (AUC) values (defined as the degree of separability measuring the accuracy of a quantitative diagnostic test) of both the QUIPP app and quantitative FFN were 0.898 (95% CI 0.850 to 0.946) and 0.902 (95% CI 0.857 to 0.946), respectively [64].

These tools are essentially early or late predictors of PTB. As early predictors, they offer an opportunity to initiate surveillance and treatment in the form of progesterone or cervical suture. In the acute setting, they are also useful triaging tools that can guide hospital admission and the initiation and provision of supportive obstetric or neonatal care for a likely PTB, such as the administration of steroids for fetal lung maturity (Figure 2, Boxes 1 and 2). However, despite these tools and pathways, PTB continues to affect 15 million pregnancies every year; and with a complex, multifactorial etiology, precision-based medicine appears to confer a new hope in this area [8].

## 3. Precision-Based Medicine

Precision-based medicine, previously known as personalized medicine, is an emerging approach for disease prevention and management. It is a concept in which the diagnosis and treatment of conditions is based on identifying which approaches will be most effective for certain patients by assessing, for example, genetic, environmental, and lifestyle factors. In essence, this approach is much more "tailored to the individual characteristics of each patient" [66]. Most medical conditions are complex and heterogenous, and they are equally influenced by the affected individual and their environment. Thus, the population approach, which has been the norm in medicine for decades, is shifting to be more individualized. Examples of precision-based medicine in current clinical practice include the cross-matching of blood and phenotyping of tumors for targeted therapy. In the case of matching donated blood to a patient requiring a transfusion, screening for red cell antibodies and red cell phenotype helps to prevent adverse reactions and improves the chances of donor compatibility. It is also used in the treatment of receptor-positive breast cancer and stem cell engineering. This paper will discuss the role of precision-based medicine for PTB prevention and avenues of treatment. We will explore current research and development in biomarker discovery, specifically in the fields of infection and inflammation, and genomics, as well as the use of artificial intelligence.

### 3.1. Biomarkers

There are a number of biomarkers (Figure 4) associated with the pathophysiology of PTB that have the potential to be used for clinical diagnostics (Figure 5).

### 3.1.1. Bacterial Biomarkers

1. Targeting the vaginal microbiome

Infections related to the reproductive tract have been investigated as an etiology of PTB. However, a causal link between sexually transmitted infections (STI) and PTB is uncertain. Fetal and neonatal concerns regarding common STIs (such as chlamydia and gonorrhea) and less common infections (such as syphilis, HIV) are mostly aimed at vertical transmission. However, HIV infection is associated with an increased risk of PTB, and as discussed earlier, immune dysregulation associated with infection is likely to contribute to this finding [67]. A large population-based retrospective cohort study of 14,373,023 pregnant women concluded

that the odds ratio (measure of association between exposure and outcome) for PTB relating to chlamydia was 1.11, relating to gonorrhea 1.17 and relating to syphilis 1.06 compared with pregnant women without infection [68]. This study formed the basis of targeted prevention prior to and screening during the antenatal period. In the UK, HIV and syphilis tests are offered as part of the antenatal booking appointment through the NHS [69,70]. However, other sexually transmissible diseases such as chlamydia and gonorrhea are not routinely screened. Nevertheless, screening is recommended by NICE for 15- to 24-year-old men and women who are sexually active [71]. Following this important work, there has been much speculation around the potential for targeted interventions around STI screening, resulting in personalized treatment, though critics have identified a lack of data linking the impact of treatment to the prevention of PTB [72]. One of the techniques in development is a novel multiplex real-time PCR melting curve assay method for the simultaneous detection of *Chlamydia trachomatis*, *Neisseria gonorrhoeae*, *Mycoplasma genitalium*, *Trichomonas vaginalis*, *Mycoplasma hominis*, *Ureaplasma urealyticum*, *Ureaplasma parvum* and *herpes simplex* virus [73]. Results have shown a sensitivity of 91.06–100% and a specificity of 99.14–100% for diagnosing these infections using vaginal swabs. Importantly, the test has excellent repeatability and is easy to use, which makes this a highly acceptable test in the clinical setting.

With the most recognized etiology of PTB being closely linked to infection, precision-based methods to identify and manage infective agents are at the forefront of development in this area. The healthy vaginal microbiome during pregnancy consists of high levels of Lactobacilli. Different species of the genus Lactobacillus have also been found to differ by race and ethnicity; Caucasian women were found to have relatively higher *L. crispatus* and *L. gasseri* colonization, whereas Hispanic and African American women had higher *L. iners* [74]. Several studies have found that the presence of *Lactobacillus crispatus* within the vaginal microbiome carries an even lower risk of PTB [75].

However, a paucity of lactobacilli is linked to a predisposition for vaginal anaerobic colonization. Anaerobic colonization facilitates concomitant colonization by bacterial vaginosis, which is known to be linked to a higher risk of genital infection, pelvic inflammatory disease and human papillomavirus (HPV) [76]. Studies have also found that high species diversity in the vagina is linked to infection and PTB [77–79]. Furthermore, the bacterial vaginosis-associated organisms *G. vaginalis*, *A. vaginae* and *Veillonellaceae* bacterium in the vaginal microbiome may be associated with an increased risk of PTB [80]. Furthermore, the link between vaginal and uterine bacterial colonization has been demonstrated in non-pregnant women, where dysbiosis of both compartments with shared species is seen during inflammatory uterine pathology such as endometritis [81]. This suggests a vaginal origin for uterine infections that could be a good target for diagnostics.

2. Targeting amniotic fluid microbial colonization

Though maternal systemic infections are not a major cause of PTB in the post-antibiotic era [82], almost 60% of cases of spontaneous PTL are thought to be due to a microbial invasion of the amniotic cavity (MIAC)—with a likely source being the vagina. Invading organisms will often include bacteria that are known to be associated with PTB in the literature, such as Mycoplasma and Ureaplasma species [83,84]. The seminal work by the ORACLE collaborative in the 2000s investigated the use of erythromycin or co-amoxiclav for PTL. The antibiotics were given to pregnant women presenting with PTL with intact membranes and/or preterm ruptured membranes before 37 weeks. They showed a reduction in maternal infection and short-term improvement in neonatal morbidity, but not in perinatal mortality [85–87]. Importantly, in the intact membrane group, antibiotics did not improve neonatal outcomes or delay birth, and the use of co-amoxiclav was associated with necrotizing enterocolitis in the neonate. The mixed effectiveness in this study may be due to the non-specific nature of the antibiotics used. Consequently, clinical practice is to give erythromycin for preterm ruptured membranes only. Moreover, when followed up at the 7-year mark, children were not found to have any significant health effects relating to antibiotic use with preterm rupture of membranes. However, if membranes were intact at onset of labor, both erythromycin and co-amoxiclav were found to be associated with an increased chance of functional impairment

and a 1% higher risk of cerebral palsy [85,86,88]. New methods to detect intra-amniotic infection and targeted treatment would be a better model of care.

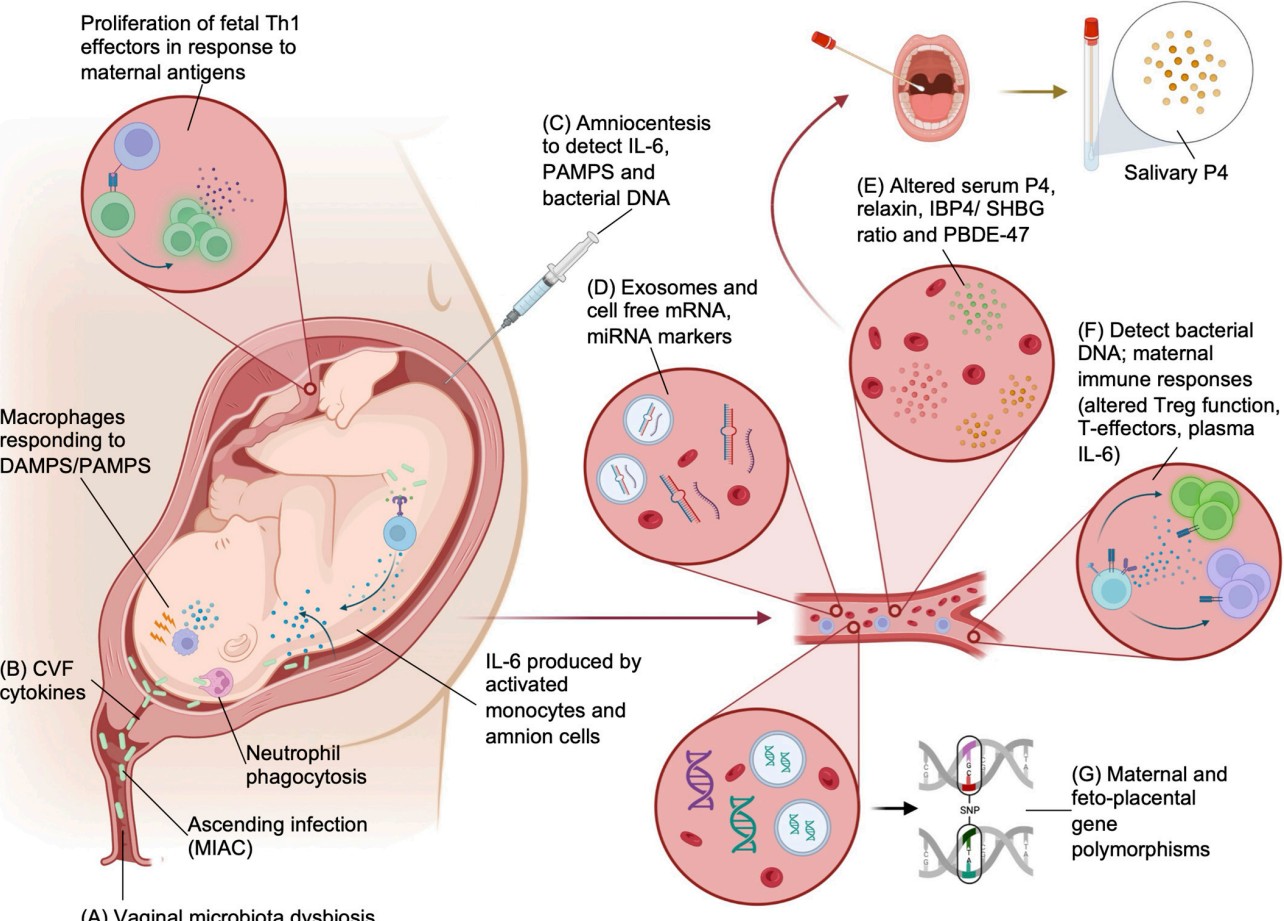

**Figure 4.** Biomarkers that can be used for the prediction, detection and guidance of the treatment of preterm birth. (**A**,**B**) Vaginal microbiome and cervicovaginal fluid (CVF) can be used to detect dysbiosis and cervical cytokines that predispose to PTB. (**C**) Amniocentesis can be used to sample amniotic fluid to detect both intra-amniotic bacteria and products of the maternal immune response (IL-6 and PAMPs/DAMPs). Ascending organisms (MIAC) into the amniotic cavity may induce an immune response including phagocytosis by neutrophils and PAMP and DAMP generation that will activate macrophages, causing pro-inflammatory cytokine release (IL-6). Activated intracavity monocytes and amnion cells will also produce IL-6. Infective PTB is also associated with greater fetal exposure to maternal antigens and proliferation of proinflammatory fetal T cells. (**D**) Several candidate cell-free RNA signatures (mRNA and miRNA) have been discovered in the plasma of pregnant women that are associated with PTB. (**E**) Measurable hormonal biomarkers that have been shown to have immunomodulatory functions and myometrial quiescence include progesterone (P4), relaxin, IBP4/SHBG ratio and PBDE-47. These can be detected in maternal plasma and in some instances, in maternal saliva. (**F**) Systemic infection and immune responses associated with infection and PTB can be detected from peripheral blood. Methods to detect infections include traditional NAATs that are rapid and highly sensitive but lack isolation of AMR and 16S rRNA. Whole-genome metagenomics give better coverage, and new methods such as multiplex PCR are rapid and suitable as a point-of-care test. Maternal immune response can be detected with fluorescence-based immune profiling to detect a shift in immune-suppressive phenotypes and plasma cytokines. (**G**) Maternal and fetal polymorphisms in genes associated with components of potential etiological pathways of PTB have been explored and are detectable in maternal blood. Created with BioRender.com—accessed 13 December 2023.

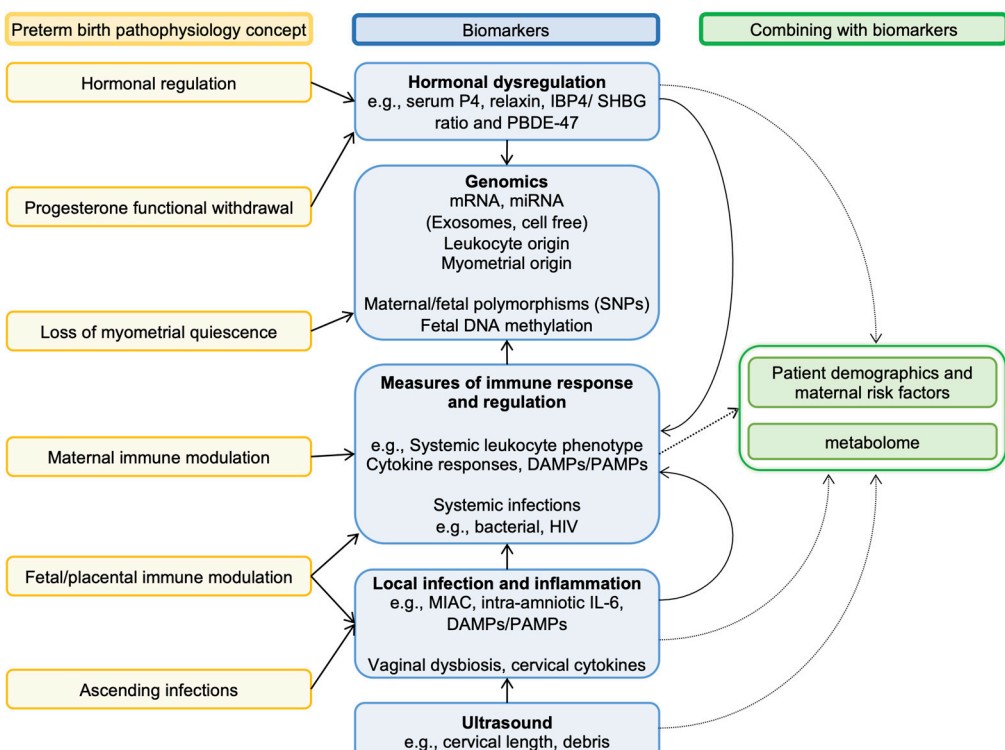

**Figure 5.** Linking key pathophysiological concepts of preterm birth with biomarkers. Biomarkers are linked to key concepts of the pathologies thought to be causes of preterm birth. Most pathologies are multifactorial and can often progress to affect other systems/compartments. An example is vaginal bacteria, which can invade the amniotic cavity (i.e., MIAC), leading to systemic infection and disrupting maternal immune modulation of the feto-placental unit. This is illustrated in the diagram by how various biomarkers are connected. Several researchers have further refined the utility of many biomarkers by combining these with maternal/fetal risk factors as well as associated metabolites (metabolome).

Mendz et al. conducted a systematic review of the literature, identifying 13 observational studies that investigated intra-amniotic infection leading to PTB. By combining the total number of PTB cases (761), they found 349 cases of MIAC associated with PTB. They then reported the prevalence of different species of bacteria found in the intra-amniotic cavity. These included Actinobacteria (25), Firmicutes (343), Fusobacteria (71), Bacteroidetes (20) and Proteobacteria (58). Unsurprisingly, the women were found to have more than one species upon testing [89]. However, 25% of PTBs may be due to intra-amniotic infection, most commonly from Ureaplasma species [90]. These infections often remain unnoticed, as they do not induce symptoms or signs in the woman systemically, but they have been shown to respond to maternal antibiotic treatment. Clarithromycin, for example, has been shown to reduce intra-amniotic inflammation measured by IL-6 levels and Ureaplasma bacterial DNA load in pregnant women with preterm ruptured membranes and less than 34 weeks of gestation [91]. In murine studies, some of the benefit from clarithromycin treatment is due to downregulation of alarmin-induced inflammation that occurs as part of the immune response against microbial infection [92].

Zhu et al. investigated amniotic fluid and placental samples in healthy pregnancies. They found that all amniotic fluid samples were culture-negative, and 20% of placental samples were found to be culture-positive, suggesting that bacterial colonization of the cavity can occur in healthy pregnancies [93]. Moreover, Stinson et al. used 16S rRNA methods to analyze neonatal first-pass meconium to show that neonatal gut microbes will have originated from the amniotic fluid. They also reported heterogeneous colonization between subjects, which suggests that the normal intra-amniotic microbiome is diverse [94].

Subsequently, the same group compared mid-trimester amniotic fluid samples obtained following amniocentesis between pregnant women who eventually delivered at term or preterm. They found that in 20% of the preterm cohort, the samples contained bacterial DNA, demonstrating widespread and early intrauterine colonization [82]. This suggests an association with bacterial colonization and tissue inflammation. Urushiyama et al. investigated the association between amniotic fluid and placental inflammation with bacterial colonization using 16S rDNA sequencing. They identified 11 strains of bacteria associated with chorioamnionitis and PTL. These included *Ureaplasma parvum*, *Streptococcus agalactiae*, *Gardnerella vaginalis*, *Streptococcus anginosus*, *Sneathia sanguinegens*, *Eikenella corrodens*, *Prevotella bivia*, *Lactobacillus jensenii*, *Bacteroides fragilis*, *Porphyromonas endodontalis* and *Mycoplasma hominis*. In addition, they discovered that the bacteria found in placentas with very mild infection or no infection were completely different from those found in patients with fulminant chorioamnionitis [84].

Another possible method of detecting clinically relevant and pregnancy-specific infections has been described by Pustotina et al. and uses ultrasound scanning. The presence of amniotic fluid "sludge" at the internal cervical os on transvaginal ultrasound scan was found to be associated with maternal intra-amniotic infection. The authors compared the use of antibiotics in women in whom sludge was identified versus no treatment. If no intervention was given, the women had a PTB rate of 46%, but this decreased significantly once antibiotic therapy was administered [95,96]. Targeting MIAC etiology for PTL using point-of-care diagnostics designed around precision-based medicine and then offering early treatment could significantly delay the latency to birth and potentially prevent some instances of PTB.

3. Targeting maternal infection

A less-invasive method than amniocentesis to detect maternal-fetal infections, and perhaps even intra-amniotic organisms would be ideal. However, this would require detection of organisms without sampling the compartment with the highest concentration of organisms to detect. Currently, PCR-based technology that uses multiplex is already available in clinical practice, that provides rapid detection of organisms and antimicrobial genes from a predefined panel tailored to the sample source [97,98]. For more comprehensive characterization, novel methods for next-generation sequencing have been developed that can be used to detect organisms, their resistance genes and enables profiling of host immune responses [99]. Techniques such as real-time nanopore sequencing as a method of rapid identification of bacteria, antibiotic resistance genes and plasmids in blood cultures have been shown to enable correct use of antibiotics within 4 h of a positive blood culture [100]. Other methods include cell-free metagenomic sequencing, which provides a way to identify organisms and their resistance genes without the issue of host DNA contamination but also enables the profiling of host tissue injury as a measure of host response [101]. In a series of intensive care patients, compared to traditional blood cultures, microbial cell-free DNA (mcfDNA) showed better sensitivity but not specificity (68.1 and 63.2% versus 40.4 and 82.8% for mcfDNA and blood cultures, respectively). It included the detection of viruses and provided a much wider pathogen spectrum, with better identification of less systemically invasive pathogens [102]. However, in these examples the clinical samples were still obtained from the source of infection such as blood (for sepsis) or tracheal aspirates/broncho-lavage specimens (for respiratory infections). In a cohort of patients with suspected chorioamnionitis, Witt et al. attempted to detect fetal exposure to microorganisms using whole-genome sequencing [103]. They showed that mcfDNA signatures in the umbilical cord and maternal plasma during preterm and term delivery differed between patients with and without chorioamnionitis demonstrating a less invasive method of detecting intra-amniotic infection. By specifically examining organisms associated with early-onset neonatal sepsis and/or PTB in the literature, they showed species enrichment in the presence of clinical infection [103]. These studies suggest that similar techniques can be used to develop diagnostics for clinical (e.g., chorioamnionitis) and

subclinical maternal-fetal infections (e.g., infective PTL), and enable the provision of early treatment, which would improve PTB-related outcomes for both mother and the neonate.

## 4. Targeting indicators of maternal immune response

The phenotype of circulating T cells has promise as a measurable biomarker or series of biomarkers for the prediction of PTB. The mechanism underpinning this work is twofold, representing the concept of the loss of immune tolerance to the semi-allogenic fetus by the host immune system prior to labor, as well as the maternal immune response to intra-amniotic infection driving PTL. For example, Frascoli et al. found fetal T helper 1 (Th1) cells from preterm infants exhibiting a proinflammatory response to maternal antigens within the maternal circulation, which were not found in term infants [104]. Moreover, in murine work, these T cells stimulated uterine myometrial cell contractility. Other groups have observed a decline in proportions or a shift in regulatory T cell immune suppression to favor proinflammatory effector T cell immune responses with the onset of labor [105,106]. This suggests that changes in the maternal immune function with labor onset may be a part of normal pregnancy physiology in preparation for birth.

Interleukin-6 (IL-6) has been linked to risk of PTB [107–109]. Automated electro-chemiluminescence immunoassay methods have been developed to detect and confirm the presence of IL-6 in intra-amniotic and cervical fluid samples of pregnant women who are delivering prematurely [110]. Park et al. went on to investigate plasma IL-6 through a retrospective cohort study but found that although less sensitive and specific than amniotic fluid IL-6, plasma IL-6 levels are equivalent in sensitivity to C-reactive protein (CRP) for detecting intra-amniotic infection predictive of imminent PTB [111]. Early work is being conducted to better identify possible proteins that reduce IL-6-mediated immune response as a precursor to pharmacological solutions to delay preterm birth [112]. Though tradition-ally processed through enzyme-linked immunosorbent assay (ELISA), Chaemsaithong et al. investigated the performance of a commercially available lateral flow-based immunoassay using amniotic fluid that can deliver results within twenty minutes, and they showed that this had a specificity of 96% and a sensitivity of 97% to detect intra-amniotic infection with a concentration of IL-6 of 745 pg/mL or more [113]. However, similar methods can be used as a point-of-care test for plasma cytokines.

Other biomarkers related to the acute diagnosis and management of PTB currently being researched include complement through the immune response pathway, serum thiol/disulphide to predict threatened preterm labor (tPTL) within the following seven days and procalcitonin as an indicator of histological chorioamnionitis in women with premature rupture of membranes after steroids [114–120]. Splichal et al. investigated the in vivo expression and release of damage- and pathogen-associated molecular patterns (DAMPs and PAMPs) in a pig model of intra-amniotic infection using *E coli*. DAMPs and PAMPs bind to pattern recognition receptors on immune cells and tissue to induce an inflammatory immune response. Their DAMP of interest was the High Mobility Group Box 1 (HMGB1) nuclear protein [121]. Amniotic membrane expression of HMGB1 was downregulated during intra-amniotic infection, whereas its release in amniotic fluid was increased. This work was reproduced in murine models with similar conclusions [122]. Research on HMGB1 in human cord blood samples, amniotic fluid and placental histology has also demonstrated a correlation with PTB outcomes [123]. Analyzing these immunological factors as surrogate biomarkers offers a novel approach for the early identification of PTL.

### 3.1.2. Hormonal Markers

Abuelghar et al. conducted a prospective observational study to compare serial salivary progesterone concentrations with serial cervical length ultrasound measurements to predict spontaneous PTL after 24 weeks [124]. For all participants (134 women), these measurements were taken at 26 weeks and repeated 3–4 weeks later and grouped based on the gestation of birth for analysis (early PTB defined as 28–34 weeks, late PTB as 34–37 weeks and >37 weeks as term births). Interestingly, the initial measurement and the rate of reduction of both salivary progesterone and cervical lengths were greater in

women who delivered preterm. As a tool to predict early PTB, they found that low salivary progesterone levels were comparable to cervical length. The area under the curve (AUC) for the first visit cervical length and salivary progesterone was 0.937 and 0.984, respectively, and for the second visit, the values were 0.908 and 0.976. Serial differences in only cervical length or salivary progesterone were not statistically significant, but when combined, the performance was much better.

Similarly, elevated serum relaxin has been associated with PTB [125]. This is believed to be mediated by the effect of relaxin to increase leukocyte numbers, stimulating their functions (including adhesion and migration) during pregnancy [126,127]. Impacted cells include uterine NK cells, macrophages and neutrophils, which are believed to stimulate uterine contraction [126]. In non-human primates, the hormone relaxin, produced by the corpus luteum, has been shown to support endometrial decidualization and endometrial parenchymal remodeling through upregulation of NK cells and the inhibition of progesterone receptor isoforms PRA and PRB and estrogen receptor (ER)$\alpha$ [128]. A decline in endometrial PRA/B and Er$\alpha$ expression has been shown in the secretory phase of the menstrual cycle. Relaxin also maintains endometrial connective tissue integrity through the inhibition of pro-matrix metalloproteinase-1 (pro-MMP-1), which usually causes extracellular matrix degradation [128]. However, relaxin has been shown to have differential effects on MMPs in the cervix, where it causes degradation of the connective tissue and ripening of the cervix. It is through this mechanism that it is believed to increase the risk of PTB [129]. Other potential serum hormonal biomarkers include the IBP4/SHBG ratio and PBDE-47, which have been shown to be predictive of PTB weeks or months ahead of time, enabling monitoring, planning and preparation [118].

### 3.1.3. Genomic Markers

There is increasing work being conducted to consider the use of precision-based medicine through genomics. Genomics can be used to predict PTB through the recognition of DNA/RNA and gene polymorphisms and through the identification of fetoplacental genes.

1. Targeting cell-free RNA

As a biomarker, cell-free RNA may represent disease activity and are extremely stable in the circulation, which makes them particularly suitable for clinical use. Zhou et al. at Michigan State University have been building on the concept of predicting PTB through the identification of maternal serum biomarkers in their nested case-control study. They used the bioinformatic tools miRWalk and STarMirDB to search for miRNA transcripts that target the *EBF1* gene, which is a transcription factor that is important during lymphopoiesis, and to regulate B and T cell lineage [130]. This was chosen based on a previous work by the same group in which they screened six genes (EBF1, EEFSEC, AGTR2, WNT4, ADCY5 and RAP2C) and their variants that have been linked to gestational duration and/or spontaneous PTB in the literature [131]. Patient samples were taken at 17–23 weeks and 27–33 weeks and subsequently grouped into PTB (<37 weeks) or term births (>37 weeks) based on the gestation of birth. They found that four potential maternal blood *EBF1*-based miRNA transcripts (*MIR4266*, *MIR1251*, *MIR601*, *MIR3612*) were significantly associated with spontaneous PTB. For samples taken in the third trimester (27–33 weeks), when combined these had a specificity of 72% and sensitivity of 81% for predicting PTL (AUC 0.82; Yoden's index 0.53, 95% CI 0.26–0.69). When combined, the performance of the four transcripts was significantly more accurate in predicting PTB when compared with single transcripts ($p < 0.0034$) [132].

Weiner et al. conducted an ex vivo study to identify differences in second-trimester cell-free RNA signatures (mRNA and miRNA) in the plasma of pregnant women who went on to deliver preterm and compared this to women who experienced birth at term [133]. This was one of the first transcriptome-wide validation studies to look at second-trimester plasma samples for predictive genomic biomarkers. The team used microarray and qRT-PCR assays and identified five RNA markers known to interact with preterm initiator

genes (myometrial in origin) that were significantly increased in women who delivered at <32 weeks of gestation [133]. To try to understand their origin, the team investigated the expression of these in placental tissue and myometrium. Their results showed that the RNAs were also overexpressed in the placenta of pregnant women who delivered at <32 weeks of gestation [133]. Moreover, these RNAs were associated with molecular pathways that have not been previously linked to PTB and suggested a disruption of myometrial quiescence that may be a precursor to PTB [133]. However, further research is required to further validate this data.

2. Targeting maternal gene polymorphisms

Gupta et al. identified different metabolite signatures in the sera of second-trimester women at risk of PTL in comparison with term controls using $^1$H nuclear magnetic resonance (NMR) metabolomics and genome-wide-screening microarray techniques [134]. More recently, Bhattacharjee et al. conducted a genome-wide association study of spontaneous PTB on 6211 women from India, and following subgroup analysis, they found five single-nucleotide polymorphisms (SNPs; rs4798499, rs2689089, rs7645913, rs10026052 and rs16238) that could help to identify women with an increased risk of early spontaneous PTB (<33 weeks of gestation) [135]. Rocha et al. used genotyping through PCRs followed by quantitative immunocytochemistry (ELISA) to identify SNPs in the Relaxin H2 (RLN2) promoter. As discussed earlier, elevated serum relaxin has been associated with PTB. The group identified one SNP, rs4742076 ($p = 0.001$), that was associated with PPROM, and one SNP, rs3758239 ($p = 0.0002$), associated with both PPROM and spontaneous PTB in Filipino patients [136].

The Preterm Birth Genome Project is an international consortium that plans to investigate the genetic predisposition to PTB. This is conducted through genome-wide association studies looking for SNPs. The aim of the project is to develop a bespoke PTB array. This approach will also enable a better understanding of PTB physiology and eventually provide a way to offer medications that are more effective for individual patients with a particular history and at a certain gestation [137].

However, this is not the only work being conducted to determine genetic risk indicators for PTB. The research team at the Liverpool Women's Hospital in the UK conducted a cohort study using blood samples from women in their second trimester to test through the UK Biobank Axiom Array for genome data and the Clariom D Human assay for transcriptome analysis. They found that a genome-wide significant SNP, rs14675645 (*ASTN1*), was associated with spontaneous PTB, whereas the microRNA-142 transcript and PPARG1-FOXP3 gene set were associated with PPROM at a gestation of 20 weeks. This work has demonstrated the potential for multi-omic biomarkers as predictive tools for PTB [134].

SNPs are also being investigated as potential markers to identify when making decisions about the use of tocolytics in certain patients. One such study discovered that the tocolytic sulindac sulphide's plasma concentration at the last measured timepoint from initial dosing was increased in patients with a specific genotype of a polymorphism of the flavin-containing mono-oxygenase 3 (FMO3) and flavin-containing mono-oxygenase 6 (FMO6) genes [138]. Those who were variant-type homozygotes of FMO3 were found to have much higher serum concentrations of sulindac sulphide for longer. This demonstrates the potential for SNPs to predict individualized pharmokinetics, and thus their importance for treatment efficacy. Furthermore, the identification of certain gene polymorphisms can also tell us whether a patient in PTB will experience adverse side effects to the B2 adrenergic agonist ritodrine [139].

3. Targeting fetoplacental genes

Li et al. from Stanford University analyzed a dataset of de novo mutations in whole blood samples obtained from 816 parent-offspring trios using whole-genome sequencing to show that PTB is associated with a significant increase in de novo mutation burden in fetal genomes [140]. Chien et al. took this further in 2019, when they studied the molecular expressions of mesenchymal stem cells in preterm infants. They used an integrative analysis

of dual-omics data to investigate 5615 commonly identified genes/proteins and found 29 genes/proteins that showed a consistent pattern of up- or downregulation in PTB. These methods could be used to identify markers that could be early predictors of PTB [141].

Another study in the United States investigated the epigenetic profiles of neonates born preterm. The authors looked at DNA methylation in cord blood samples, and through a bipartite network analysis, they found statistically significant inverse symmetrical relationships in the case and control group methylation profiles. Using a method known as Ingenuity Pathway Analysis to reveal any meaningful biological relationships, the group identified cellular senescence as an important pathway. They also discovered a strong inverse linear relationship between total methylation and gestational age that could be used in the personalized prediction of length of gestation [142].

One of the more recent breakthroughs in this area includes the development of a transcriptomic signature-based model by Ran et al. in China, which has been validated to identify women at risk of PTB within the following seven days. Using peripheral blood mRNA expression data, the team performed weighted correlation network analysis and a gene set enrichment analysis to pinpoint the key genes involved in preterm delivery. They determined that the following four core genes impacted the development of PTB: JOSD1, IDNK, ZMYM3 and IL1B. In combination, they conferred a sensitivity and specificity of 83.9% and 87.0%, respectively (with a PPV 86.6%, NPV 84.4%). This work promises a novel strategy for clinical risk assessment and management in the acute setting [143].

### 3.2. Artificial Intelligence and Technology

Artificial intelligence and technology are other methods that are being explored with regards to the prediction, diagnosis and treatment of PTB (Table 2).

### 3.2.1. Computational Modeling

There are several examples of the use of computational modeling to investigate uterine smooth muscle cell activation and contraction. Aslanidi et al. developed a computational model that recognized uterine cell activation [144]. Tong et al. developed a computational model that took 105 differential equations into account to understand uterine smooth muscle cell excitation [145]. Bastos et al. built a computational model to simulate intrauterine pressure in women in labor. It has been used as a delivery simulator to support the training of healthcare professionals, and Lobo et al. later used it to simulate the effect of oxytocin on a laboring uterus [146,147]. These early experiments formed the basis of more recent sophisticated modeling. An example of this is Goldsztejn and Nehorai's computational myofiber model, which studied both electropropagation and uterine force development on intracellular coupling and cellular excitability. Their work described the complexity of electrochemical pressure dynamics in the initiation of contractions [148].

Computational modeling has also allowed for the discovery of personalized therapeutic interventions for preventing PTB. Le et al. used computational models through rank-based pattern matching to compare the gene expression signatures of women who were at risk of PTB with the differential gene expression drug profiles in a published Connectivity Map database [149,150]. This led them to assign a reversal score to each PTB-drug pair, and they identified 13 potential compounds that were safe in pregnancy and may be effective in preventing PTB [150].

### 3.2.2. Machine Learning

This has been used to identify novel associations between risk factors, serum markers and clinical pathology with PTB. The Vanderbilt Genetics Institute led a retrospective study of the electronic health records related to 35,282 deliveries in the US and employed a machine-learning framework and boosted decision tree models (non-parametric supervised learning algorithm) to predict spontaneous PTB [151]. This model used billing codes up to 28 weeks of gestation and was able to predict spontaneous PTB with a sensitivity of 48%, which was significantly higher than a risk factor-only prediction model (sensitivity

of 35%). The machine-learning model was also able to risk-stratify deliveries based on co-morbidities and risk factors.

Research using machine learning to predict PTB has also been conducted in China using a cohort of 65,525 pregnant women. The machine-learning models used were generalized additive models (statistic modeling technique used to analyze the relationship between response and predictor variables) with penalized cubic regression spline (non-parametric method of regression modelling) to explore the non-linear association between maternal thyroid hormone (T4) and risk of PTB. The data were gathered through history taking and blood draw in antenatal clinics by midwives and obstetricians [152]. The time-to-event method and multivariable cox proportional hazard model (regression model investigating the length of time) were further applied to look more closely at the possible association between very high and low maternal T4 concentrations with the timing of PTL. Through this work, they identified a U-shaped association between maternal T4 and gestational age at delivery but found no evidence of thyroid disease being a risk factor for PTB. This suggests that though thyroid pathology may not be linked with PTB, serum-free T4 level may be a useful marker for a screening test to identify women at risk of PTB [152].

### 3.2.3. Artificial Intelligence

With the advent of artificial intelligence accelerating at pace across all industries, research groups have investigated its use to analyze datasets to develop new predictive tools for PTB. For example, several groups have used neural networks to classify uterine electrical activity into different labor phenotypes [153–155]. Chen and Xu built deep neural networks for the semi-automatic classification of preterm versus term uterine electrical activity [155]. They fed data from tocograms (instrument that records uterine muscle contraction activity) into wave and sample entropies, which extracted features and submitted them to the deep neural network. This enabled the team to classify the uterine recordings and predict PTB with a sensitivity of 98% and a specificity of 97.7%.

The ECCLIPPx prospective cohort study examined the efficacy of a new cervical probe device based on electrical impedance spectroscopy to predict PTB in asymptomatic women. The probe measured transfer impedance by applying electrical current at fourteen frequencies ranging from 76.3 Hz to 625 kHz in increments through an adjacent pair of injecting electrodes, and voltage was measured between the remaining pair of sensing electrodes. This device was able to predict PTB with a sensitivity and specificity of 80% and 85% respectively in high-risk women and 73% and 84% in asymptomatic women. Moreover, with both groups combined when electrical impedance spectroscopy was assessed at 20–22 weeks, the device was able to predict PTB with an AUC of 0.76 (95% CI, 0.71–0.81, $p < 0.0001$) [156]. This was better than cervical length USS (AUC 0.72, 95% CI, 0.66–0.76), $p < 0.001$) and FFN (AUC 0.62, 95% CI, 0.56–0.72, $p = 0.05$) alone. However, when combined their performance was significantly better (AUC, 0.79, 95% CI, 0.74–0.83, $p < 0.05$). Other studies, such as the one conducted by Tomialowicz et al., investigated bioelectric activity measured by electrohysterography rather than mechanical activity to predict PTB. This approach was preferred since bioelectric activity precedes mechanical activity in the initiation of labor. In this study, measurement of bioelectric uterine activity has been found to enable personalized tocolytic treatment for pregnant women [157]. Based on these studies, the ability to predict the earliest indication of PTL could give the managing healthcare team more time to investigate potential causes of PTL in each patient and deliver targeted treatment that could prevent or delay PTB.

**Table 2.** Summary of research in artificial intelligence and technology exploring mechanisms to diagnose, understand and delay PTB.

| Research Team | Model/Study Design | Methods | Sample Size (n) | Main Findings | Prediction/Diagnosis/Treatment of PTB |
|---|---|---|---|---|---|
| | | Computational | | | |
| Aslanidi et al., 2011 [144] | Computational/in vitro Experimental | Computational models were created from electrohysterogram data to understand the electrical activity in a pregnant uterus using data from various experiments on cells and tissues. Virtual tissue engineering of uterine tissue was developed using in vivo magnetic resonance imaging (MRI) and ex vivo diffusion tensor magnetic resonance imaging (DTI) to create these models, which aim to predict how the uterus contracts during labor. Similar tools are used for heart contraction modeling, where there is better data. | n/a | 1. Application of virtual tissue engineering to understand electrical activity in myometrial tissue. 2. Use of MRI and DTI to simulate a 3D model of the uterus. 3. Development of a computational model to measure electrical propagation through smooth muscle cells. | Diagnosis and treatment Using these models to study uterine activity during labor will advance understanding of the mechanisms underlying initiation and progression of labor with potential for direct diagnosis, management and treatment of PTL. |
| Tong et al., 2011 [145] | Computational/in vitro Mathematical modeling | The study employed a computational biology approach to develop a mathematical model describing the excitation-contraction (E-C) coupling of uterine smooth muscle cells (USMC). Fourteen ionic currents in USMCs were quantified using differential equations based on published and unpublished data, including maximal conductance, voltage-dependent gating variables and intracellular calcium changes. | n/a | 1. The advanced mathematical model developed is useful for researching the physiological ionic mechanisms that underpin uterine E-C coupling during labor and parturition. This is an early model that should be refined to create better predictive models of myometrial electrical activity at both the cellular and tissue levels. | Diagnosis This model can be used to investigate and predict myometrial electrogenesis at both the cellular and tissue levels, contributing to a better understanding of how to identify normal and dysfunctional labors. |
| Le et al., 2020 [150] | Computational and murine Observational: case control | The researchers used a rank-based pattern-matching approach to compare the differential gene expression signature for PTB with drug profiles in the Connectivity Map database. They assigned a reversal score to each PTB-drug pair to identify drugs with potential efficacy in preventing PTB. The drug lansoprazole was selected for further validation. | 30 | 1. The study identified 83 drugs, including lansoprazole, with significantly reversed differential gene expression compared to PTB. 2. Lansoprazole, a proton-pump inhibitor, showed a strong reversal score and a good safety profile. In an animal inflammation model using LPS, lansoprazole demonstrated a significant increase in fetal viability compared to LPS treatment alone. | Treatment This study highlights the potential of computational drug repositioning to discover compounds that could be effective in preventing PTL. |

| Research Team | Model/Study Design | Methods | Sample Size (n) | Main Findings | Prediction/Diagnosis/Treatment of PTB |
|---|---|---|---|---|---|
| Goldsztejn and Nehorai, 2020 [148] | Computational/in vitro Experimental | The researchers employed a computational model to study the relationship between electrical propagation, force development, intercellular coupling and cellular excitability in the myofiber. | n/a | 1. The myometrium becomes significantly more excitable and contractile as the uterus remodels in preparation for delivery. 2. Abnormal remodeling can lead to preterm birth, slow progress of labor and failure to initiate labor. 3. The study finds that intercellular coupling is a determinant of the conduction velocity in the myometrium. 4. Intercellular coupling by itself does not regulate the force development in the myometrium. 5. Current pharmaceutical treatments for uterine contraction disorders primarily target increased intercellular coupling and cellular excitability, both outcomes of remodeling. | Treatment This study used a computational model to understand how cellular functions like intercellular coupling and cellular excitability impact tissue properties such as electrical propagation and force development in the myometrium. This understanding is the start of developing advanced treatments for PTB. |
| | | | | Machine Learning | |
| Abraham et al., 2022 [151] | Machine learning Experimental | Machine-learning models were developed using billing codes and known risk factors from EHRs of 35,282 deliveries. The models were used to predict preterm birth risk at different gestational ages and compared with models based on known risk factors. The patterns learned by the model were examined to stratify deliveries into interpretable groups. | 35,282 | 1. Machine-learning models based on billing codes alone showed promising results in predicting preterm birth risk and outperformed models based on known risk factors. 2. The approach was demonstrated to be portable and accurate on an independent cohort from a different healthcare system. 3. These models outperformed models that were trained using established risk factors. | Prediction This study has shown that machine-learning models could be employed to identify patients at risk of preterm birth from as early as the booking appointment. |

**Table 2.** *Cont.*

| Research Team | Model/Study Design | Methods | Sample Size (n) | Main Findings | Prediction/Diagnosis/Treatment of PTB |
|---|---|---|---|---|---|
| Zhou et al., 2022 [152] | Human Observational: cohort | A hospital-based cohort study using generalized additive models with penalized cubic regression spline was used to explore the non-linear association between maternal thyroid hormone (FT4) levels and the risk of PTD, including its subtypes. The time-to-event method and multivariable Cox proportional hazard model were applied to analyze the association of abnormally high and low maternal FT4 concentrations with the timing of PTD. | 65,565 | 1. The study revealed a U-shaped dose-dependent relationship between maternal FT4 levels in the first trimester and the risk of PTD. 2. Both low and high maternal FT4 levels were associated with an increased risk of PTD. 3. Isolated hypothyroxinemia was associated with spontaneous PTD, and overt hyperthyroidism was related to iatrogenic PTD. | Prediction This study suggests that measurement of maternal FT4 levels could be used in the prediction and prevention of preterm birth. |
| | | | Artificial Intelligence | | |
| Maner et al., 2007 [153] | Human Observational: case control | Uterine EMG signals were measured trans-abdominally using surface electrodes. Bursts of elevated uterine EMG corresponding to uterine contractions were quantified using power spectrum peak frequency, burst duration, number of bursts per unit time and total burst activity. Artificial neural networks (ANN) were used to classify patients into labor and non-labor groups, and the percentage of correctly categorized patients was calculated. | 185 | 1. The study demonstrates that ANN, when applied to uterine EMG data, can effectively classify pregnant patients into term labor, preterm labor, term non-labor and preterm non-labor categories. 2. The accuracy of classification was high, suggesting the potential utility of ANNs in objectively categorizing pregnant patients based on their labor status. | Diagnosis Artificial neural networks, when used in conjunction with uterine EMG data, can effectively categorize patients into term or preterm and laboring or non-laboring, thus enabling diagnosis of PTL. |
| Most et al., 2008 [154] | Human Observational: cohort | Electrical uterine myography (EUM) was measured on 87 pregnant women with gestational age less than 35 weeks. The researchers developed an index score (1–5) for predicting preterm delivery (PTD) within 14 days of the test based on the period between contractions, power of contraction peaks and movement of the center of electrical activity (RMS). The EUM index score was compared with fetal fibronectin (fFN) and cervical length (CL) to assess its predictive ability. | 87 | 1. Patients who delivered within 14 days from testing showed a higher EUM index and mean RMS. which suggests that electrical uterine myography (EUM) was able to identify preterm birth. 2. When EUM data were combined with either CL or fFN, the predictive accuracy increased. 3. Other factors such as gestational age at the time of testing, body mass index, fFN and CL did not significantly impact the risk of PTD. | Diagnosis Measuring myometrial electrical activity may enhance the identification of patients in true premature labor and rule out those who are not in labor. |

**Table 2.** *Cont.*

| Research Team | Model/Study Design | Methods | Sample Size (n) | Main Findings | Prediction/Diagnosis/Treatment of PTB |
|---|---|---|---|---|---|
| Chen, L. and H. Xu. 2007 [155] | Human Observational: case-control | The study aimed to investigate the potential of a sparse autoencoder-based deep neural network (SAE-based DNN) in predicting preterm birth using ElectroHysteroGram (EHG) and Tocography (TOCO) signals, which are real-time and non-invasive technologies. The deep neural network (DNN) model was used to measure the bursts of uterine contraction intervals and non-contraction intervals (dummy intervals) from 26 recordings of the TPEHGT DS database that were manually segmented. The SSAE network was used to learn high-level features from these raw features through unsupervised learning. The proposed method was evaluated using 10-fold cross-validation and four performance indicators. | 26 | 1. The deep neural network achieved high performance in predicting preterm birth, with sensitivity of 98.2%, specificity of 97.74% and accuracy of 97.9% on the TPEHGT DS database. 2. The DNN outperformed comparison models, such as deep belief networks (DBN) and hierarchical extreme learning machine (H-ELM). | Diagnosis This method shows promise for the semi-automatic identification of term and preterm uterine recordings, offering a non-invasive approach for enhancing the diagnosis of PTL. |
| Anumba et al., 2021 [156] | Human Observational: cohort | The study aimed to evaluate the predictive performance of a cervical probe device based on electrical impedance spectroscopy (EIS) for preterm birth (PTB). The goal was to compare this method with the existing prediction methods—transvaginal ultrasound (TVS) cervical length (CL) measurement and fetal fibronectin (FFN)—in asymptomatic women during the mid-trimester. Multivariate linear and non-linear logistic regression analyses were used to assess the associations of cervical EIS, TVS-CL and FFN with spontaneous PTB before 37 weeks and before 32 weeks. Areas under the receiver operating characteristics curves (AUC) were calculated to compare the predictive performance of the parameters individually and in combination. | 365 | 1. Cervical EIS assessment in the mid-trimester predicted spontaneous PTB before 37 weeks with an AUC of 0.76, outperforming TVS-CL (AUC 0.72) and FFN (AUC 0.62). 2. Combining all three assessments (EIS, TVS-CL and FFN) improved the prediction of spontaneous PTB before 37 weeks (AUC 0.79). 3. Incorporating a history of spontaneous PTB further improved the accuracy of prediction before 37 weeks (AUC 0.83) and before 32 weeks (AUC 0.86). | Prediction The mid-trimester assessment of the cervix using EIS is effective in predicting spontaneous PTB. |

**Table 2.** *Cont.*

| Research Team | Model/Study Design | Methods | Sample Size (n) | Main Findings | Prediction/Diagnosis/Treatment of PTB |
|---|---|---|---|---|---|
| Tomialowicz et al., 2021 [157] | Human Observational: prospective cohort | Forty-five pregnant women with pregnancies ranging from 24 to 36 weeks of gestation and typical clinical symptoms of threatened preterm delivery were treated with tocolytic therapy. Bioelectric activity was recorded using electrohysterography simultaneously with mechanical activity recorded using tocography. | 45 | 1. Measurement of bioelectric activity was more sensitive in detecting uterine activity compared to mechanical activity.<br>2. Despite the use of tocolysis and the absence of symptoms of PTL, increased bioelectric activity in the uterine muscle was detected. In contrast, tocography did not show significant contractions.<br>3. Bioelectric activity may precede the occurrence of mechanical activity of the uterus. | Diagnosis<br>This study suggests bioelectric activity might be an early indicator, potentially preceding the mechanical activity of the uterus and therefore a mechanism for earlier diagnosis of PTL. |

The academic consensus is that the milestones required to achieve artificially intelligent device-led prediction include early identification, clinical risk stratification, miniaturization, specialized critical care protocols, a regulatory path and a strategy and platform to translate technology to the bedside. What may aid in the development of such artificial intelligence tools is the use of organic in vitro models to enable learning of uterine and fetal physiology—for example, data generated on fetal physiological support using an extracorporeal VV-ECLS artificial placenta (AP) or an AV-ECLS artificial womb (AW) [158]. Similarly, the team at Tohoku University in partnership with the University of Western Australia have come up with an ex vivo uterine environment [159] which was developed to enable lung maturation and organ development by the Children's Hospital of Philadelphia for their EXTEND project (EXTra-uterine Environment for Neonatal Development) [160]. These systems have been designed to provide ex utero support for preterm neonates, with the aim of mimicking the in utero fetal environment. Though these devices have not been developed to predict or delay PTB and are currently too immature for application in the clinical setting, there is the potential to use a similar approach to model PTL and use the complex datasets generated to teach artificial intelligence tools. An example is the analysis of uterine electrical activity using neural networks, for prompt diagnosis and evaluation of tocolytic interventions. The application of artificial intelligence for predicting and managing PTB has a great deal of promise. However, it is important to appreciate the current limitations of using artificial intelligence in this context. One of the primary constraints is data quality and availability due to artificial intelligence's limited access to comprehensive scientific databases, leading to a dependence on small and varied types of data sources. Part of the problem is the complexity of factors affecting PTB and its etiology, which give rise to numerous routes of investigation. This issue poses a significant challenge in terms of knowledge acquisition and application. In addition to this, there is the issue of algorithm bias, whereby the outputs generated may favor certain patient groups and make it difficult to apply the findings to diverse populations. Therefore, integrating the results of these into generalizable artificial intelligence models is difficult. For a potential clinical tool, any artificial intelligence model proposed will need to be reliable, accurate and consistent. Only then can such a model be used in a clinical workflow. Consequently, a cautious approach is recommended when considering the use of artificial intelligence in the context of PTB.

*3.3. Combining Methods*

So far, this paper has explored the effect of individual markers and techniques in the identification, diagnosis and management of PTL. More recently, researchers have been combining different techniques to further improve their predictive accuracy for PTL and PTB. An example of this is the use of multiple biomarkers rather than a single one. Goldenberg et al. proposed using a combination of serum biomarkers (defensins, $\alpha$-fetoprotein, granulocyte colony-stimulating factor), cervical length ultrasound and vaginal FFN with a history of previous spontaneous PTB. Low levels of defensins have been linked with PTB, chorioamnionitis and neonatal sepsis [161], $\alpha$-fetoprotein is recognized as a protein released after the rupture of membranes and is a part of the ROM-plus point-of-care diagnostic test [54], and higher levels of serum granulocyte colony-stimulating factor have been associated with PTB through possible leucocyte-mediated interference with placental function [162]. Two positive test results provided a sensitivity and specificity of 78.8% and 62%, respectively, with an odds ratio of 6.0 for predicting PTB [163]. In a nested case control study on human pregnant participants performed by Huang et al., the team screened the utility of 35 maternal serum cytokines. They found that the AUC was 0.546 and 0.559 for TNF$\alpha$ and TRAIL, respectively, in a single-cytokine model. Higher levels of TNF$\alpha$ and TRAIL have been linked to placental dysfunction and intra-amniotic infections relating to PTB [164]. However, this improved significantly with multiple cytokines (IL-8, IL-9, IL-10, IL-13, IL-18, Eotaxin, IP-10, TNF-$\alpha$, VEGF, TRAIL, MCP-3, TNF-$\beta$, and HGF) to an AUC of 0.642 [165]. To refine their model further, the team stratified by BMI, gestational age at sample collection and gestational age at PTB. With this combined approach, the

multiple cytokines had an AUC of 0.694, 0.637 and 0.879 in groups defined by BMI < 18.5, BMI 18.5–23.9 and BMI ≥ 24, respectively, which demonstrated a clear improvement in the accuracy of diagnosis.

Kindschuh et al. took 232 vaginal samples from women in their second trimester and used 16S ribosomal RNA amplicon sequencing, with the results put through machine-learning models to predict spontaneous PTB from vaginal fluid [166]. The results showed a strong association between the vaginal microbiome and metabolome and identified metabolites that when used together were able to predict PTB. Overall, the microbiome populations were well clustered in community state types, with disparity among racial groups. Among metabolomes, Ethyl β-glucopyranoside (ethyl glucoside; $p$ = 0.00019); tartrate ($p$ = 0.00048); and diethanolamine (DEA; $p < 10^{-10}$) and Ethylenediaminetetraacetic acid (EDTA) ($p$ = 0.00016) were all raised in spontaneous PTB and are of an exogenous source. Though mechanisms of action for these metabolites have not yet been investigated, they have been linked to cosmetic and hygiene products, which suggests that common environmental agents may also influence PTB risk. Furthermore, it was found that choline, an essential nutrient and a precursor of the amnio acid betaine, was negatively associated with spontaneous PTB ($p$ = 0.007). The additional use of machine learning to process and combine the different factors through boosted decision trees combining metabolomic and microbiome data was able to improve the predictive performance of the parameters with good accuracy (AUC of 0.78).

Chiu et al. combined maternal risk factors, uterine artery doppler indices, serum PLGF, pregnancy-associated plasma protein-A (PAPP-A) and β-human chorionic gonadotropin (β-hCG) to develop a predictive model [58]. Data from 9298 singleton pregnancies were analyzed in this retrospective study. The authors found that screening by a combination of maternal risk factors, PAPP-A and PLGF achieved better performance in predicting spontaneous PTB at <37 weeks (AUC, 0.630 vs. 0.555; detection rate, 24.8% vs. 16.6% at a specificity of 90%; $p \leq 0.0001$) than using maternal risk factors alone. This indicates that a combined method of prediction with risk factors, dopplers and serum biomarkers is more accurate than predicting risk of PTB or PPROM with maternal risk factors alone. However, further research is needed to optimize biomarker combinations before they can be used in clinical practice.

Much like the QUiPP app (where quantitative fFN and cervical lengths were used) [64], Radan et al. combined cervico-vaginal fluid testing for placental α-macroglobulin-1 (PAMG-1, Partosure®) with ultrasound measurements of cervical length in patients with symptoms of PTL to predict PTB within 14 days [167]. The sensitivity for Partosure® alone is 83% in detecting 1 ng/mL or more of PAMG-1, and it has a specificity of 95% [167]. The authors found that the combined test provided a higher specificity (97–98%) and positive predictive value (100%) when a cervical length of <15 mm was measured than when individual techniques were used (PPV 71.4% for Partosure® alone). This study of 126 women showed promising results, and the authors plan to conduct a prospective observational trial to evaluate the utility of a combined testing method compared to using placental α-macroglobulin-1 alone.

These studies illustrate how diagnostic accuracy is improved through combined techniques over individual methods. Moreover, the QUiPP app is a good example of how this can be developed into a widely used clinical tool. It may be that the multifactorial etiology of PTB can potentially be tackled through combining the different diagnostic methods depending on the clinical history and presentation to improve their efficacy.

## 4. Precision-Based Medicine to Direct Treatments

### 4.1. Prevention Treatment

There are many ways in which precision-based medicine can improve clinical care related to PTB. For example, the current gold standard for Group B Streptococcus (GBS) identification is the culture of vaginal swab samples, which take 48 h to yield results that can influence patient care. More recently, lateral flow assays for detection of GBS have be-

come available with good specificity, and they provide a point-of-care test [168]. These use a nucleic acid amplification process (NAAT) that is highly specific and sensitive, but they do not provide additional information that can be clinically useful such as antimicrobial resistance or sensitivity [169]. The mechanism by which NAATs are used in lateral flow assays is the identification of small amounts of DNA or RNA samples—in this case, the GBS cAMP factor gene. This is unlike 16S rRNA, metagenomics and culture methods. However, new NAATs and next-generation sequencing (including multiplex-based PCR) provide a method of point-of-care testing that can augment the traditional diagnostic NAAT or a lateral flow. These methods do not require time-consuming enrichment cultures and isolation of pure cultures but are able to use non-purified polymicrobial clinical samples [170,171]. With results available within 30 min or less, these point-of-care assays enable accurate identification of GBS in the outpatient, acute or intrapartum clinical setting, enabling prompt management to improve perinatal outcomes. There are several advantages of using new NAAT and next-generation sequencing methods over traditional 16S rRNA, metagenomics and culture methods for diagnostic testing. These include the rapidity of results to enable prompt management. These tests have a high sensitivity and specificity and so can be relied upon to make decisions. They aid in quantitative analysis and have an ease of use, requiring less clinical time and having improved antibiotic stewardship. The disadvantages of these point-of-care tests are their cost (more costly than cultures), availability and the technical limitation of not having the sensitivity information that cultures provide [149].

An example of current point-of-care testing is the GBS3 trial, where researchers are using a NAAT system to diagnose GBS and then provide prophylactic antibiotics for laboring women [172]. However, the use of a rapid point-of-care test can also help to direct vaccines. Globally, an estimated 20 million pregnant women were thought to have been colonized with GBS in 2020 (230,000 neonatal infections), and approximately 3.5% of all PTBs in the same year were likely associated with GBS (518,000 births) [173]. Immunization against GBS would have a significant impact in preventing both neonatal sepsis and PTB. However, a GBS vaccine needs to not only provide immunity against GBS colonization but also induce a sufficient maternal humoral immune response to protect the mother and neonate from infection. Robust and sustained vaccine immunogenicity will protect against ascending infection and transfer seroprotective antibodies across the placenta to protect the neonate. Fortunately, the immunogenicity of current vaccine candidates appears to meet these criteria, and initial results are promising [174].

Research on GBS prevention in murine models through a recombinant alpha-like protein subunit vaccine been effective, as vaccinated mice had IgG titers versus controls, and specifically, though vaccination did not eliminate GBS during ascending infection in pregnancy, vaccinated mice experienced fewer in utero fetal deaths [175]. The current global position on GBS vaccines is at a Phase II trial level. A hexavalent Ia, Ib, II, III, IV and V glycoconjugate vaccine has completed Phase I clinical trials, showed positive results for healthy adults and was cleared to progress to Phase II trials [176]. The vaccine is designed to provide coverage for 98% of GBS isolates that are known to affect the fetus in utero. Furthermore, it has been shown to be sufficiently immunogenic and is able to induce opsonophagocytosis to clear both GBS colonization and infection, and it appears to be acceptable to patients [174]. However, further trials are required to understand immunogenicity and tolerability and to assess for potential side effects prior to release in the clinical setting. The specific timing of administering a GBS vaccine during pregnancy is not certain; however, evidence indicates that for maximum levels of anti-GBS antibodies to be transferred to the fetus, the most impactful time of administration would be during the late second to early third trimester. Key considerations in making this decision include an understanding of the duration of immunity, safeguarding the fetal and maternal immune response, ensuring acceptability and tolerance, vaccine safety profile, understanding requirements around storage and administration that may exacerbate inequalities in health, epidemiological factors, and integration with existing antenatal schedules.

It is estimated through modeling that once deployed globally, a GBS vaccine would prevent 185,000 PTBs [177]. This would be through the administration of a one-dose vaccine, which would cost approximately USD 1.7 billion to deploy. However, any associated healthcare burden from delivering the vaccine would be balanced through the prevention neonatal infections and the subsequent effect on quality of life, and. In fact, a GBS vaccine would save USD 385 million in healthcare costs and confer a global net monetary benefit of between USD 1.1 and 17 billion dollars [157]. The success of this vaccine at a global level could support the development of other common causative agents, thus preventing further spontaneous PTBs and fetal deaths. Indeed, GBS is a good example, but the same treatment can be extrapolated to other organisms identified as causative for PTB. What is clear is that research and development of diagnostic methods such as NAATs, PCR and next-generation sequencing to identify causative organisms and the use of specific vaccines have a promising future.

### 4.2. Acute Treatment

Very little clinical options exist for pregnant women in this setting. With a high prevalence of microbial invasion of the amniotic cavity driving spontaneous PTL, novel methods to identify infections (as discussed earlier), profile maternal immune responses and provide targeted treatment may be beneficial. In a non-human primate model of PTL using intrauterine-administered *Escherichia coli* (*E. coli*), Cappelletti et al. have shown that giving systemic antibiotics 24 h after injecting *E. coli* is effective at eradicating bacteria but does not resolve choriodecidual or amnion inflammation. In their model, PTL was only prevented in in 25% of cases [178]. During infective PTB, the IL-1 signaling pathway drives intrauterine inflammation and chorio-decidua neutrophil recruitment [179]. However, it is important to appreciate that in this model of PTB, the *E. coli* used was a highly uropathogenic strain (UTI89), injected at a concentration of $10^6$ CFU/mL. This is known to induce an exaggerated proinflammatory response in vitro, with raised expression of cytokines IL-1b (21-fold) and IL-8 (6-fold) by choriodecidual and amniotic tissue, respectively [180,181]. For the purposes of establishing an animal model of infective PTB, this will have the desired effect of inducing tissue inflammation and pro-contractile myometrial responses. In humans, commonly identified strains of *E. coli* are usually from phylogroups B2 and D, which are less uropathogenic than UTI89 and are more likely to colonize in the genital or intestinal tract [182]. Moreover, though *E. coli*, Streptococcus species and Haemophilus influenzae are prevalent in preterm early neonatal sepsis, the commonly associated microorganisms in subclinical intra-amniotic infection are Mycoplasmas (*Ureaplasma* spp. and *Mycoplasma* spp.) [183,184]. Consequently, there are several limitations to translating the findings of Cappelletti et al. to clinical practice. Firstly, the biological differences between non-human primates and humans can range from differing immune responses, metabolic pathways, and genetics, which may influence the effectiveness of antibiotics (dosage and pharmacodynamic effects), including side effect profiles. Secondly, inducing PTL in this model may not accurately replicate essential pathophysiologic mechanisms of PTL in humans. Finally, in clinical practice, it is unlikely that a singular pathogenic strain of bacteria will be identified as a causative agent.

Other groups have had some success in human studies. Lee et al. used amniocentesis to identify intraamniotic infection in 314 women with tPTL. They showed that a combination of ceftriaxone, clarithromycin and metronidazole was able to double the median antibiotics-to-delivery interval, with lower rates of histologic chorioamnionitis (23 versus 12 days, $p < 0.05$) [185]. In contrast, women treated in the absence of infection and/or inflammation did not see a difference in latency to birth, with many delivering preterm. Other research groups have since directly investigated amniotic fluid infection in women with PTL using cytokine ELISA and 16S rDNA. Combs et al. conducted a prospective study, conducting amniocentesis on 305 pregnant women with PTL. They measured IL-6 levels in the amniotic fluid and stratified groups based on IL-6 quantities into negative, mild, moderate and severe inflammation and cultured the fluid for bacterial growth. Their results

showed that the degree of inflammation with or without the presence of bacteria directly correlated with PTB and poor neonatal outcomes [68]. Yoon et al. conducted a similar study with post-antibiotic follow-up amniocentesis, which also identified intra-amniotic inflammation by detection of IL-6. The team was able to measure IL-6 levels before and after antibiotic treatment and observed resolution of inflammation in 32% of patients. They calculated the median amniocentesis-to-delivery interval was significantly longer among women who received the combination of ceftriaxone, clarithromycin and metronidazole than among those who did not (11.4 days vs. 3.1 days; $p = 0.04$) [163]. Through these studies, researchers have demonstrated that IL-6 is a good surrogate marker for intra-amniotic infection (infective PTL) and a predictor of poor outcomes, and that maternal antibiotics are effective at eradicating intra-amniotic bacteria [83,186].

As an adjunct to antibiotic treatment, immune modulators provide a method to reduce inflammation to treat or delay PTB. For example, in vivo murine work using dimethyl-formamide (DMF) decreases the rate of spontaneous abortion ($p < 0.0001$) and PTB in LPS-induced pregnant mice ($p < 0.0001$) [187]. In humans, for the treatment of multiple sclerosis, DMF has been shown to result in a shift in memory T cell responses from effector memory to naïve, thus reducing inflammation, but without dampening the ability to mount effective recall and humoral responses when exposed to antigens [188,189]. Studies looking at the effect of DMF in PTB have not yet been conducted.

Other potential candidates for adjunctive treatment include immune-active molecules. Suff et al. investigated whether the enrichment of cervical mucosa with a defensin molecule can provide immune protection against ascending infections. In a murine model, they used a viral vector to induce expression of AMP from the human β-defensin-3 (HBD3) gene (including a luciferase transgene for bioluminescence) in the cervical mucosa of pregnant mice and then administered a pathogenic *E. coli* strain (K1 A192PP) into the vagina [190]. HBD3 encodes for a protein that functions as an antimicrobial agent and chemoattractant [191]. In treated mice, they were able to demonstrate improved *E. coli* killing in vitro using vaginal lavage specimens and reduced ascent of bacteria into the uterine cavity. Specifically, their results showed a significant reduction in uterine bioluminescence in the HBD3 gene-treated mice when observed 24 h after *E. coli* infection was introduced and when compared to controls. The use of these adjuncts could be beneficial with regards to delaying PTB long enough for the clinical team to put a management plan in place for safer delivery.

Currently, the main proposed interventions in the acute period are antibiotics and adjunct. This is because very little can be done immediately in an acute presentation of PTB to prevent and delay physiologically established labor. There is clearly a long way to go in this area. However, using other methods of precision-based medicine for prediction and detection may guide management earlier in pregnancies to reduce the incidence of acute presentations and the associated perinatal sequelae.

### 5. Conclusions

In conclusion, there are certainly many different avenues in development with regards to the prediction, acute diagnosis and treatment of PTB. Work on the identification of infective agents is most developed, with bacterial infections and host immune responses being identified and assessed using serum, amniotic fluid and vaginal fluid samples. However, researchers have identified a number of techniques to assess a range of biomarkers that have the potential to further personalize the early detection and prediction spontaneous PTB as well as develop point-of-care diagnosis and targeted treatment. Genomics is currently still in its infancy within this context. Further work involving the validation of genes and transcripts and integration with other omics techniques needs to be conducted. Work incorporating artificial intelligence and the use of technology is also in its early stages and requires further development, validation and clinical modeling before application to any clinical setting. Though these techniques exist, there will always be a cohort that requires acute management in the clinical setting, and this remains the most challenging group of

patients. It is highly likely that combining methods to develop precise diagnostic tools will enable the best form of personalized care for the individual.

**Author Contributions:** Individual contributions as follows: conceptualization, N.M.S.; methodology, H.K.; validation, J.M. and N.S.; formal analysis, N.M.S. and H.K.; resources, N.M.S. and J.M.; data curation, H.K., N.M.S. and L.Y.L.; writing—original draft preparation, H.K. and N.M.S.; writing— review and editing, N.M.S., H.K., N.S., J.M. and L.Y.L.; supervision, N.M.S.; project administration, N.M.S. All authors have read and agreed to the published version of the manuscript.

**Funding:** This research received no external funding.

**Data Availability Statement:** No new data were created or analyzed in this study. Data sharing is not applicable to this article.

**Acknowledgments:** Acknowledgements are made to the Department of Metabolism, Digestion and Reproduction, Imperial College London, and the University of Central Lancashire School of Medicine for their administrative support. Infrastructure support was provided by the NIHR Imperial Biomedical Research Centre and the NIHR Imperial Clinical Research Facility. The views expressed are those of the authors and not necessarily those of the NIHR or the Department of Health and Social Care. Graphical abstract and figures created with Biorender.com—accessed between 1 November and 15 December 2023.

**Conflicts of Interest:** The authors declare no conflicts of interest.

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
