# Peer review of "Does Precision-Based Medicine Hold the Promise of a New Approach to Predicting and Treating Spontaneous Preterm Birth?"

_2673-8937, doi:10.3390/ijtm4010002_

Round 1
Reviewer 1 Report
Comments and Suggestions for Authors
Thank you for allowing me to review the article titled “Does precision-based medicine hold the promise of a new approach to predicting and treating spontaneous preterm birth?” (ijtm-2723895). The purpose of this review is to examine the role of precision-based medicine in predicting, preventing, and treating preterm birth.
Comments:
In the abstract, it's essential to outline the background justifying the study, its objectives, methodology, key results, and conclusion. The current abstract only discusses the topic's importance and objective. It's vital for the readers of this journal to grasp an overview of the approach and the main ideas this review contributes. Therefore, I suggest it be rewritten in line with the standard structure of an abstract.
The introduction should be expanded as it currently uses only two references and addresses a topic of significant importance. I recommend elaborating on the working hypothesis and stating the objective more clearly and precisely.
On line 61, could you please clarify the meaning of “MBRRACE”?
On line 63, you mention that a crucial aspect of this work is “PERIprem”, yet this isn't referred to in the objectives.
The review is well-structured into thematic sections, which aids in better understanding the subject.
In line 667, and again in line 719, Lee et al. are mentioned without providing a bibliographic reference.
Table two is intriguing, but it should indicate the sample size and how these studies were conducted.
Overall, I believe this topic is of great current importance. However, literature reviews are increasingly being used to update healthcare professionals' knowledge. Thus, when seeking to update information, various reviews are conducted. It's therefore crucial to specify the type of review undertaken. I understand it to be a comprehensive review of the subject, but it's for the authors to define the design. It's also essential to state the period the review covers, the sources of articles used, and the inclusion and exclusion criteria. I also believe that artificial intelligence represents a new methodology that we should be open to in the future. However, I'm unclear how it is applied in this study and would appreciate clarification on these aspects.
Author Response
Many thanks for your feedback. Please see attached cover letter.

Reviewer 2 Report
Comments and Suggestions for Authors
- This is a well written and informative paper. Please come across with the following comments
-
Introduction: The introduction provides a clear overview of the significance of preterm birth (PTB) as a global health concern, emphasizing its impact on neonatal morbidity and mortality. The mention of the World Health Organization's recognition of PTB as a public health priority lends credibility to the importance of the topic. The identification of the UK's statistics and the challenges faced in managing PTB sets the stage for the discussion on the need for precision-based medicine.
-
Preterm Birth: The categorization of PTB into sub-categories is informative, giving readers a context for the severity of cases. The discussion on the associated morbidities and mortality rates linked to the gestational age at birth is supported by relevant statistics. The improvement in survival rates due to interventions is highlighted, and the mention of initiatives like PERIprem adds a practical dimension to the narrative.
-
Risk Factors and Pathophysiology: The section on risk factors is well-structured, categorizing them as modifiable and non-modifiable. The inclusion of a table adds visual clarity to the information. The exploration of multifactorial pathophysiology, including inflammatory processes, neuroendocrine mechanisms, and various maternal conditions, is thorough. The connection between chronic systemic infections like HIV and PTB is discussed, integrating global statistics to underscore the impact.
-
Multifactorial Etiology: The detailed exploration of various causes, including cervical insufficiency, uteroplacental insufficiency, and the role of mechanical factors, enhances the understanding of the multifaceted nature of PTB. The integration of molecular and immunological explanations provides a comprehensive perspective.
-
Clinical Guidelines: The reference to guidance from the National Institute of Clinical Excellence (NICE) and WHO adds practical relevance to the review. The consideration of multidisciplinary team discussions and patient preferences in clinical decision-making reflects a holistic approach.
-
Overall Impression: The paper is well-written, providing a comprehensive overview of PTB, its complexities, and the potential role of precision-based medicine. The incorporation of statistics, examples, and guidance enhances the credibility and practical application of the information. The paper could benefit from further clarification in some areas, such as the potential conflicting reports on the association between protease inhibitors and PTB in HIV-positive pregnant women.
-
Recommendation: The paper is informative and relevant, making a valuable contribution to the understanding of PTB. Addressing minor clarification points and ensuring up-to-date references would enhance its overall impact.
Section 3.2: Current Clinical Tools to Predict Preterm Birth
Strengths:
-
Comprehensive Overview: The section provides a thorough overview of current clinical tools, including ultrasound, serum biomarkers, vaginal biomarkers, and a clinical decision-making app.
-
Evidence-Based Information: The inclusion of specific biomarkers such as Placental Growth Factor (PlGF) and their approval for use by NHS adds credibility to the information presented.
-
In-Depth Analysis: The discussion of the sensitivity, specificity, and predictive values of PlGF using the Triage MeterPro point-of-care analyzer enhances the depth of the analysis.
-
Integration of Technologies: The integration of different technologies, such as transvaginal ultrasound and quantitative point-of-care tests, demonstrates a comprehensive approach to PTB prediction.
-
Application in Clinical Practice: The mention of the QUIPP app developed by researchers from King's College London and its high predictive values adds practical relevance to the discussion.
-
Temporal Aspect Considered: The distinction between early and late predictors of PTB, and the discussion of their applications in initiating surveillance and treatment, adds a temporal dimension to the analysis.
Suggestions for Improvement:
-
Clarity on Terminology: Consider providing a brief explanation or definition of terms like "sensitivity," "specificity," and "predictive values" for readers who may not be familiar with these concepts.
-
Visual Aid: Given the complexity of the information, incorporating a visual aid such as a table or figure summarizing the key characteristics of each clinical tool could enhance readability.
-
Consistency in Terminology: Ensure consistent use of terminology throughout the section. For instance, "Placental Growth Factor (PlGF)" is initially spelled out, but later referred to as "PlGF." Maintaining consistency is crucial for clarity.
-
Cross-Referencing: Consider cross-referencing specific studies or findings mentioned in the section. This can help readers who are interested in delving deeper into the literature.
Section 3. Precision-based Medicine
Strengths:
-
Introduction to Precision Medicine: The introduction to precision-based medicine and its emerging role in disease prevention and management is well articulated.
-
Relevance to Current Medical Practices: The mention of precision medicine examples in current clinical practice, such as cross-matching blood and tumor phenotyping, provides context for readers.
-
Applicability to PTB: The clear connection between precision-based medicine and PTB prevention, including biomarker discovery, infection and inflammation, genomics, and artificial intelligence, offers a holistic perspective.
Suggestions for Improvement:
-
Transition Clarity: Enhance the transition between the discussion on current clinical tools and precision-based medicine to maintain a smooth flow of ideas.
-
Brief Definition: Provide a brief definition or explanation of precision-based medicine to ensure understanding among readers who may not be familiar with the term.
Section 3.1: Biomarkers
Strengths:
-
In-Depth Exploration: The section provides a comprehensive exploration of biomarkers associated with the pathophysiology of PTB, covering infection and inflammation, hormonal markers, and genomic markers.
-
Evidence-Based Information: The inclusion of studies, such as the population-based retrospective cohort study on infections related to the reproductive tract, adds robustness to the discussion.
-
Connection to PTB: The discussion on the association between vaginal microbiome, amniotic fluid microbial colonization, and maternal immune response with PTB is well-connected and contributes to a holistic understanding.
Suggestions for Improvement:
-
Organization: Consider organizing the information under subheadings to improve the section's overall organization and facilitate readers in navigating through different aspects of biomarkers.
-
Subheading Clarity: Use clear and concise subheadings to highlight different categories of biomarkers, making it easier for readers to follow the discussion.
-
Visual Representation: Consider incorporating visual aids, such as diagrams or figures, to illustrate complex relationships between biomarkers and PTB pathophysiology.
General Comments:
-
References: Ensure that all references are cited consistently and accurately. Verify the formatting and citation style to meet the specific requirements of the intended audience or publication.
-
Word Choice: Occasionally, there are long sentences and complex structures. Consider breaking down complex sentences for enhanced readability.
-
Check Terminology: Ensure that medical terminology is used consistently and accurately throughout the paper.
-
Conclusion: Consider providing a brief conclusion at the end of each subsection or a summary at the end of the entire paper to reinforce key points.
Section 3.3: Combining Methods
The section discusses the trend among researchers to combine various techniques to enhance the predictive accuracy for preterm labor (PTL) and preterm birth (PTB). The use of multiple biomarkers, imaging, and clinical history is explored as a means to achieve better sensitivity and specificity. Here's a breakdown and some suggestions for improvement:
-
Goldenberg et al's Multimodal Approach:
- The mention of Goldenberg et al's study combining serum biomarkers, cervical length ultrasound, vaginal FFN, and history of previous spontaneous PTB is informative.
- Consider providing a brief explanation of what these biomarkers signify or how they contribute individually to the prediction.
-
Huang et al's Study on Maternal Serum Cytokines:
- The study by Huang et al, using multiple maternal serum cytokines, adds valuable information.
- Clarify the significance of TNFα and TRAIL, and how the AUC values improved with the multiple cytokines model.
-
Kindschuh et al's Approach with Vaginal Samples:
- The use of 16S ribosomal RNA amplicon sequencing by Kindschuh et al to predict spontaneous PTB is intriguing.
- Explain the relevance of identified metabolites and how machine learning improved predictive performance.
-
Chiu et al's Predictive Model:
- The combination of maternal risk factors, ultrasound indices, and serum markers by Chiu et al is discussed.
- Provide a concise summary of the key findings and improvements achieved by combining these factors.
-
Radan et al's Combined Testing:
- Radan et al's combination of cervico-vaginal fluid testing and ultrasound measurements is highlighted.
- Elaborate on the improved specificity and positive predictive value compared to individual techniques.
-
Suggestions for Improvement:
- Provide a brief introduction or summary at the beginning of the section to set the context for combining methods in predicting preterm birth.
- Consider adding a concluding paragraph that synthesizes the key findings and emphasizes the overall trend and importance of combining methods.
Overall, the section provides a comprehensive overview of different studies combining methods for predicting preterm birth, but adding a bit more detail and context to each study could enhance clarity for the reader.
Section 4: Precision-based Medicine to Direct Treatments
4.1. Prevention Treatment:
The section discusses lateral flow assays for Group B Streptococcus (GBS) detection and the potential of a GBS vaccine. Here's a review and some suggestions:
-
Lateral Flow Assays:
- Clear explanation of lateral flow assays for GBS detection with good specificity.
- Suggest elaborating on why these assays are considered point-of-care tests and how they contribute to clinical management.
-
Nucleic Acid Amplification Process (NAAT):
- The mention of NAAT is relevant but could benefit from a brief explanation of its role in lateral flow assays.
- Highlight the advantages and limitations of NAAT in this context.
-
Comparisons with Other Methods:
- The comparison with 16S rRNA, metagenomics, and culture methods adds depth.
- Consider providing a concise summary of why NAATs and next-generation sequencing are preferred.
-
GBS3 Trial and Vaccine Development:
- The GBS3 trial using NAAT for GBS diagnosis and subsequent prophylactic antibiotics is well-explained.
- The discussion on the potential impact of a GBS vaccine globally is informative.
- Clarify the significance of the estimated prevention of 185,000 preterm births through modeling.
-
Vaccine Development Progress:
- The progress of GBS vaccine development through various trial phases is well-documented.
- Include a brief explanation of the hexavalent Ia, Ib, II, III, IV, and V glyco-conjugate vaccine and its positive results in Phase I trials.
-
Timing of Vaccine Administration:
- The discussion on the timing of GBS vaccine administration is insightful.
- Consider briefly mentioning the challenges or considerations in determining the optimal timing.
-
Conclusion of Prevention Treatment:
- A concluding sentence summarizing the potential of GBS detection assays and vaccines in preventing preterm births would enhance the section.
4.2. Acute Treatment:
This part discusses limited clinical options for pregnant women facing microbial invasion of the amniotic cavity. Here's a review:
-
Non-Human Primate Model:
- The use of a non-human primate model to study the effectiveness of systemic antibiotics after E. coli injection is well-explained.
- Clarify the limitations or challenges in translating these findings to clinical practice.
-
Inflammatory Response and Pathogenic Strain:
- The discussion on the exaggerated proinflammatory response due to the highly uropathogenic strain is informative.
- Consider briefly addressing the relevance of this model to the diversity of bacterial strains in clinical scenarios.
-
Clinical Studies and Antibiotic Treatment:
- The reference to clinical studies using amniocentesis and antibiotic treatment is valuable.
- Suggest including a concise summary of key findings from these studies.
-
Immune Modulators and Other Candidates:
- The mention of immune modulators like DMF and other potential candidates provides a broader perspective.
- Consider adding a sentence summarizing the potential of these adjunctive treatments in managing preterm births.
-
Conclusion of Acute Treatment:
- Conclude the section with a brief summary, emphasizing the current challenges and potential avenues for acute treatment.
Comments on the Quality of English Language
Well written
Author Response
Many thanks for your review and detailed feedback. Please see attached cover letter.

Round 2
Reviewer 1 Report
Comments and Suggestions for Authors
I have meticulously reviewed the updated version of the manuscript titled "Does precision-based medicine hold the promise of a new approach to predicting and treating spontaneous preterm birth?" (ijtm-2723895), along with the authors' responses to the suggestions made.
The authors have indeed done a commendable job in incorporating all the requested clarifications. However, on line 90, they mention that the review was conducted over the past three years, which could be misleading for readers of a scholarly journal. It would be more precise if they specified the exact start and end years of the review period. Additionally, in Table 3, previously Table 2, there are references to works from 2007, 2008, and so forth, which do not align with the mentioned three-year period. Please clarify this discrepancy.
The choice of keywords used is also crucial to ensure transparency in the presentation of results and to facilitate future research by other authors in this field.
Furthermore, the authors must consider the significant limitation that artificial intelligence currently faces, namely its lack of access to scientific databases and reliance mainly on different types of information. This presents a considerable challenge from a knowledge perspective. Therefore, caution is advised in its use at this juncture.
